# GTA: A Geometry-Aware Attention Mechanism for Multi-View Transformers

**Takeru Miyato[1], Bernhard Jaeger[1], Max Welling[2], Andreas Geiger[1]**
[1] University of Tübingen, Tübingen AI Center  [2] University of Amsterdam

## Abstract

As transformers are equivariant to the permutation of input tokens, encoding the positional information of tokens is necessary for many tasks. However, since existing positional encoding schemes have been initially designed for NLP tasks, their suitability for vision tasks, which typically exhibit different structural properties in their data, is questionable. We argue that existing positional encoding schemes are suboptimal for 3D vision tasks, as they do not respect their underlying 3D geometric structure. Based on this hypothesis, we propose a geometry-aware attention mechanism that encodes the geometric structure of tokens as *relative transformation* determined by the geometric relationship between queries and key-value pairs. By evaluating on multiple novel view synthesis (NVS) datasets in the sparse wide-baseline multi-view setting, we show that our attention, called *Geometric Transform Attention (GTA)*, improves learning efficiency and performance of state-of-the-art transformer-based NVS models without any additional learned parameters and only minor computational overhead.

## 1 Introduction

The transformer model (Vaswani et al., 2017), which is composed of a stack of permutation symmetric layers, processes input tokens as a *set* and lacks direct awareness of the tokens' structural information. Consequently, transformer models are not solely perceptible to the structures of input tokens, such as word order in NLP or 2D positions of image pixels or patches in image processing.

A common way to make transformers position-aware is through vector embeddings: in NLP, a typical way is to transform the position values of the word tokens into embedding vectors to be added to input tokens or attention weights (Vaswani et al., 2017; Shaw et al., 2018). While initially designed for NLP, these positional encoding techniques are widely used for 2D and 3D vision tasks today (Wang et al., 2018; Dosovitskiy et al., 2021; Sajjadi et al., 2022b; Du et al., 2023).

Here, a natural question arises: "Are existing encoding schemes suitable for tasks with very different geometric structures?". Consider for example 3D vision tasks using multi-view images paired with camera transformations. The 3D Euclidean symmetry behind multi-view images is a more intricate structure than the 1D sequence of words. With the typical vector embedding approach, the model is tasked with uncovering useful camera poses embedded in the tokens and consequently struggles to understand the effect of non-commutative Euclidean transformations.

Our aim is to seek a principled way to incorporate the geometrical structure of the tokens into the transformer. To this end, we introduce a method that encodes the token relationships as transformations directly within the attention mechanism. More specifically, we exploit the relative transformation determined by the geometric relation between the query and the key-value tokens. We then apply those transformations to the key-value pairs, which allows the model to compute QKV attention in an aligned coordinate space.

We evaluate the proposed attention mechanism on several novel view synthesis (NVS) tasks with *sparse and wide-baseline* multi-view settings, which are particularly hard tasks where a model needs to learn *strong 3D geometric priors* from multiple training scenes. We show that existing positional encoding schemes are suboptimal and that our geometric-aware attention, named *geometric transform attention (GTA)*, significantly improves learning efficiency and performance of state-of-the-art transformer-based NVS models, just by replacing the existing positional encodings with GTA.

---

Correspondence to `takeru.miyato@gmail.com`. Code: https://github.com/autonomousvision/gta

## 2 RELATED WORK

Given token features $X \in \mathbb{R}^{n \times d}$, the attention layer's outputs $O \in \mathbb{R}^{n \times d}$ are computed as follows:

$$O := \text{Attn}(Q, K, V) = \text{softmax}(QK^{\text{T}})V, \tag{1}$$

where $Q, K, V = XW^Q, XW^K, XW^V \in \mathbb{R}^{n \times d}, W^{\{Q,K,V\}} \in \mathbb{R}^{d \times d}$, and $(n, d)$ is the number of tokens and channel dimensions. We omit the scale factor inside the softmax function for simplicity. The output in Eq. (1) is invariant to the permutation of the key-value vector indices. To break this permutation symmetry, we explicitly encode positional information into the transformer, which is called positional encoding (PE). The original transformer (Vaswani et al., 2017) incorporates positional information by adding embeddings to all input tokens. This *absolute positional encoding* (APE) scheme has the following form:

$$\text{softmax}\left((Q + \gamma(\mathbf{P})W^Q)(K + \gamma(\mathbf{P})W^K)^{\text{T}}\right)\left(V + \gamma(\mathbf{P})W^V\right), \tag{2}$$

where $\mathbf{P}$ denotes the positional attributes of the tokens $X$ and $\gamma$ is a PE function. From here, a bold symbol signifies that the corresponding variable consists of a list of elements. $\gamma$ is typically the sinusoidal function, which transforms position values into Fourier features with multiple frequencies. Shaw et al. (2018) proposes an alternative PE method, encoding the relative distance between each pair of query and key-value tokens as biases added to each component of the attention operation:

$$\text{softmax}\left(QK^{\text{T}} + \gamma_{\text{rel}}(\mathbf{P})\right)\left(V + \gamma'_{\text{rel}}(\mathbf{P})\right), \tag{3}$$

where $\gamma_{\text{rel}}(\mathbf{P}) \in \mathbb{R}^{n \times n}$ and $\gamma'_{\text{rel}}(\mathbf{P}) \in \mathbb{R}^{n \times d}$ are the bias terms that depend on the distance between tokens. This encoding scheme is called *relative positional encoding* (RPE) and ensures that the embeddings do not rely on the sequence length, with the aim of improving length generalization.

Following the success in NLP, transformers have demonstrated their efficacy on various image-based computer vision tasks (Wang et al., 2018; Ramachandran et al., 2019; Carion et al., 2020; Dosovitskiy et al., 2021; Ranftl et al., 2021; Romero et al., 2020; Wu et al., 2021; Chitta et al., 2022). Those works use variants of APE or RPE applied to 2D positional information to make the model aware of 2D image structure. Implementation details vary across studies. Besides 2D-vision, there has been a surge of application of transformer-based models to 3D-vision (Wang et al., 2021a; Liu et al., 2022; Kulhánek et al., 2022; Sajjadi et al., 2022b; Watson et al., 2023; Varma et al., 2023; Xu et al., 2023; Shao et al., 2023; Venkat et al., 2023; Du et al., 2023; Liu et al., 2023a).

Various PE schemes have been proposed in 3D vision, mostly relying on APE- or RPE-based encodings. In NVS Kulhánek et al. (2022); Watson et al. (2023); Du et al. (2023) embed the camera extrinsic information by adding linearly transformed, flattened camera extrinsic matrices to the tokens. In Sajjadi et al. (2022b); Safin et al. (2023), camera extrinsic and intrinsic information is encoded through ray embeddings that are added or concatenated to tokens. Venkat et al. (2023) also uses ray information and biases the attention matrix by the ray distance computed from ray information linked to each pair of query and key tokens. An additional challenge in 3D detection and segmentation is that the output is typically in an orthographic camera grid, differing from the perspective camera inputs. Additionally, sparse attention (Zhu et al., 2021) is often required because high resolution feature grids (Lin et al., 2017) are used. Wang et al. (2021b); Li et al. (2022) use learnable PE for the queries and no PE for keys and values. Peng et al. (2023) find that using standard learnable PE for each camera does not improve performance when using deformable attention. Liu et al. (2022; 2023b) do add PE to keys and values by generating 3D points at multiple depths for each pixel and adding the points to the image features after encoding them with an MLP. Zhou & Krähenbühl (2022) learn positional embeddings using camera parameters and apply them to the queries and keys in a way that mimics the relationship between camera and target world coordinates. Shu et al. (2023) improves performance by using available depths to link image tokens with their 3D positions. Besides APE and RPE approaches, Hong et al. (2023); Zou et al. (2023); Wang et al. (2023) modulate tokens by FiLM-based approach (Perez et al., 2018), where they element-wise multiply tokens with features computed from camera transformation.

In point cloud transformers, Yu et al. (2021a) uses APE to encode 3D positions of point clouds. Qin et al. (2022) uses an RPE-based attention mechanism, using the distance or angular difference between tokens as geometric information. Epipolar-based sampling techniques are used to sample geometrically relevant tokens of input views in attention layers (He et al., 2020; Suhail et al., 2022; Saha et al., 2022; Varma et al., 2023; Du et al., 2023), where key and value tokens are sampled along an epipolar line determined by the camera parameters between a target view and an input view.

## 3 GEOMETRIC ENCODING BY RELATIVE TRANSFORMATION

In this work, we focus on novel view synthesis (NVS), which is a fundamental task in 3D-vision. The NVS task is to predict an image from a novel viewpoint, given a set of context views of a scene and their viewpoint information represented as $4 \times 4$ extrinsic matrices, each of which maps 3D points in world coordinates to the respective points in camera coordinates. NVS tasks require the model to understand the scene geometry directly from raw image inputs.

The main problem in existing encoding schemes of the camera transformation is that they do not respect the geometric structure of the Euclidean transformations. In Eq. (2) and Eq. (3), the embedding is added to each token or to the attention matrix. However, the geometry behind multi-view images is governed by Euclidean symmetry. When the viewpoint changes, the change of the object's pose in the camera coordinates is computed based on the corresponding camera transformation.

Our proposed method incorporates geometric transformations directly into the transformer's attention mechanism through a *relative transformation* of the QKV features. Specifically, each key-value token is transformed by a relative transformation that is determined by the geometric attributes between query and key-value tokens. This can be viewed as a coordinate system alignment, which has an analogy in geometric processing in computer vision: when comparing two sets of points each represented in a different camera coordinate space, we move one of the sets using a relative transformation $cc'^{-1}$ to obtain all points represented in the same coordinate space. Here, $c$ and $c'$ are the extrinsics of the respective point sets. Our attention performs this coordinate alignment within the *attention feature space*. This alignment allows the model not only to compare query and key vectors in the same reference coordinate space, but also to perform the addition of the attention output at the residual path in the aligned local coordinates of each token due to the value vector's transformation.

This direct application of the transformations to the attention features shares its philosophy with the classic transforming autoencoder (Hinton et al., 2011; Cohen & Welling, 2014; Worrall et al., 2017; Rhodin et al., 2018; Falorsi et al., 2018; Chen et al., 2019; Dupont et al., 2020), capsule neural networks (Sabour et al., 2017; Hinton et al., 2018), and equivariant representation learning models (Park et al., 2022; Miyato et al., 2022; Koyama et al., 2023). In these works, geometric information is provided as a transformation applied to latent variables of neural networks. Suppose $\Phi(x)$ is an encoded feature, where $\Phi$ is a neural network, $x$ is an input feature, and $\mathcal{M}$ is an associated transformation (e.g. rotation). Then the pair $(\Phi(x), \mathcal{M})$ is identified with $\mathcal{M}\Phi(x)$. We integrate this feature transformation into the attention to break its permutation symmetry.

**Group and representation:** We briefly introduce the notion of a *group* and a *representation* because we describe our proposed attention through the language of group theory, which handles different geometric structures in a unified manner, such as camera transformations and image positions. In short, a group $G$ with its element $g$, is an associative set that is closed under multiplication, has the identity element and each element has an inverse. E.g. the set of camera transformations satisfies the axiom of a group and is called *special Euclidean group*: $SE(3)$. A (real) *representation* is a function $\rho : G \rightarrow GL_d(\mathbb{R})$ such that $\rho(g)\rho(g') = \rho(gg')$ for any $g, g' \in G$. The property $\rho(g)\rho(g') = \rho(gg')$ is called *homomorphism*. Here, $GL_d(\mathbb{R})$ denotes the set of $d \times d$ invertible real-valued matrices. We denote by $\rho_g := \rho(g) \in \mathbb{R}^{d \times d}$ a representation of $g$. A simple choice for the representation $\rho_g$ for $g \in SE(3)$ is a $4 \times 4$ rigid transformation matrix $\left[\begin{smallmatrix} R & T \\ 0 & 1 \end{smallmatrix}\right] \in \mathbb{R}^{4 \times 4}$ where $R \in \mathbb{R}^{3 \times 3}$ is a 3D rotation and $T \in \mathbb{R}^{3 \times 1}$ is a 3D translation. A block concatenation of multiple group representations is also a representation. What representation to use is the user's choice. We will present different design choices of $\rho$ for several NVS applications in Section 3.1, 3.2 and A.3.2.

### 3.1 GEOMETRIC TRANSFORM ATTENTION

Suppose that we have token features $X \in \mathbb{R}^{n \times d}$ and a list of geometric attributes $\mathbf{g} = [g_1, \ldots, g_n]$, where $g_i$ is an $i$-th token's geometric attribute represented as a group element. For example, each $X_i \in \mathbb{R}^d$ corresponds to a patch feature, and $g_i$ corresponds to a camera transformation and an image patch position. Given a representation $\rho$ and $Q, K, V = XW^Q, XW^K, XW^V \in \mathbb{R}^{n \times d}$, we define our geometry-aware attention given query $Q_i \in \mathbb{R}^d$ by:

$$O_i = \sum_{j}^{n} \frac{\exp(Q_i^{\mathrm{T}}(\rho_{g_i g_j^{-1}} K_j))}{\sum_{j'=1}^{n} \exp(Q_i^{\mathrm{T}}(\rho_{g_i g_{j'}^{-1}} K_{j'}))} (\rho_{g_i g_j^{-1}} V_j), \tag{4}$$

Using the homomorphism property $\rho_{g_i g_j^{-1}} = \rho_{g_i} \rho_{g_j^{-1}}$, the above equation can be transformed into

$$O_i = \rho_{g_i} \sum_{j}^{n} \frac{\exp((\rho_{g_i}^{\mathrm{T}} Q_i)^{\mathrm{T}} (\rho_{g_j^{-1}} K_j))}{\sum_{j'=1}^{n} \exp((\rho_{g_i}^{\mathrm{T}} Q_i)^{\mathrm{T}} (\rho_{g_{j'}^{-1}} K_{j'}))} (\rho_{g_j^{-1}} V_j). \tag{5}$$

Note that the latter expression is computationally and memory-wise more efficient, requiring computation and storage of $n^2$ values of each $(\rho_{g_i g_j^{-1}} K_j, \rho_{g_i g_j^{-1}} V_j)$ in Eq. (4) versus only $n$ values for $(\rho_{g_i}^{\mathrm{T}} Q_i, \rho_{g_j}^{-1} K_j, \rho_{g_j}^{-1} V_j)$ and $\rho_{g_i} \hat{O}_i$ in Eq. (5), where $\hat{O}_i$ is the output of the leftmost sum.

Eq. (5), given all queries $Q$, can be compactly rewritten in an implementation-friendly form:

$$O = \mathbf{P_g} \odot \mathrm{Attn} \left( \mathbf{P_g}^{\mathrm{T}} \odot Q, \mathbf{P_g}^{-1} \odot K, \mathbf{P_g}^{-1} \odot V \right), \tag{6}$$

where $\mathbf{P_g}$ denotes a list of representations for different tokens: $\mathbf{P_g} := [\rho_{g_1}, \ldots, \rho_{g_n}]$, and "$\odot$" denotes token-wise matrix multiplication: $\mathbf{P_g} \odot K = [\rho_{g_1} K_1 \cdots \rho_{g_n} K_n]^{\mathrm{T}} \in \mathbb{R}^{n \times d}$. Also, the transpose $^{\mathrm{T}}$ and the inverse $^{-1}$ operate element-wise on $\mathbf{P_g}$ (e.g., $\mathbf{P_g}^{\mathrm{T}} := [\rho_{g_1}^{\mathrm{T}}, \ldots, \rho_{g_n}^{\mathrm{T}}]$). We call the attention mechanism in Eq. (6) *geometric transform attention (GTA)* and show the diagram of (6) in Fig. 1. Note that the additional computation of GTA is smaller than the QKV attention and the MLP in the transformer when constructing $\rho_g$ from a set of small matrices, which we will detail in Section 3.2 and in Appendix A.

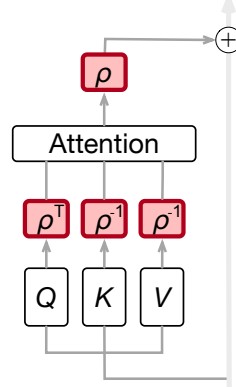

Fig. 1: **GTA mechanism.** $\rho^{-1}$ and $\rho^{\mathrm{T}}$ together take $Q, K$ and $V$ to a shared coordinate space, and the $\rho$ gets the attention output back to each token's coordinate space.

**A simple NVS experiment:** We first demonstrate that GTA improves learning as compared to APE and RPE in a simplified NVS experiment. We construct a setting where only camera rotations are relevant to show that the complexity of $\rho_g$ can be adapted to the problem complexity. A single empty scene surrounded by an enclosing sphere whose texture is shown in Fig. 2 left is considered. All cameras are placed in the center of the scene where they can be rotated but not translated. Each scene consists of 8 context images with 32x32 pixel resolution rendered with a pinhole camera model. The camera poses are chosen by randomly sampling camera rotations. We randomize the global coordinate system by setting it to the first input image. This increases the difficulty of the task and is similar to standard NVS tasks, where the global origin may be placed anywhere in the scene. The goal is to render a target view given its camera extrinsic and a set of context images.

We employ a transformer-based encoder-decoder architecture shown on the right of Fig. 2. Camera extrinsics in this experiment form the 3D rotation group: $SO(3)$. We choose $\rho_g$ to be a block concatenation of the camera rotation matrix:

$$\rho_{g_i} := \underbrace{R_i \oplus \cdots \oplus R_i}_{d/3 \text{ times}}, \tag{7}$$

where $R_i$ is the $3 \times 3$ matrix representation of the extrinsic $g_i \in SO(3)$ linked to the $i$-th token. $A \oplus B$ denotes block-concatenation: $A \oplus B = \begin{bmatrix} A & 0 \\ 0 & B \end{bmatrix}$. Because here each $\rho_{g_i}$ is orthogonal, the transpose of $\rho_{g_i}$ becomes the inverse, thus the same transformation is applied across query, key, and value vector for each patch.

We compare this model to APE- and RPE-based transformers as baselines. For the APE-based transformer, we add each flattened rotation matrix associated with each token to each attention layer's input. Since we could not find an RPE-based method that is directly applicable to our setting with rotation matrices, we use an RPE-version of our attention where instead of multiplying the matrices with the QKV features, we apply the matrices to *biases*. More specifically, for each head, we prepare learned bias vectors $b^Q, b^K, b^V \in \mathbb{R}^9$ concatenated with each of the QKV vectors of each head and apply the representation matrix defined by $\rho(g) := R \oplus R \oplus R \in \mathbb{R}^{9 \times 9}$, only to the bias vectors. We describe this RPE-version of GTA in more detail in Appendix C.1.

Fig. 3 on the left shows that the GTA-based transformer outperforms both the APE and RPE-based transformers in terms of both training and test performance. In Fig. 3 on the right, the GTA-based transformer reconstructs the image structure better than the other PE schemes.

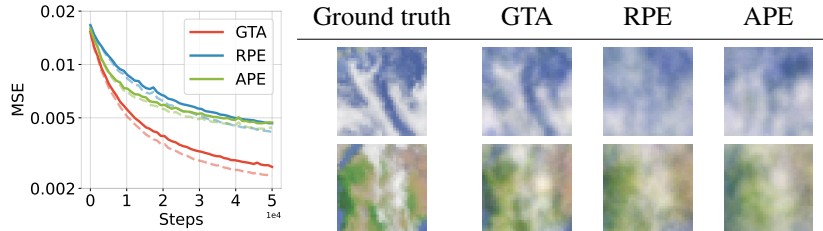

Fig. 2: **Synthetic experiment.** Left: Texture of the surrounding sphere. Right: Model architecture. The query pair consists of a learned constant value and a target extrinsic $g^*$.

Fig. 3: **Results on the synthetic dataset.** Left: The solid and dashed lines indicate test and train errors. Right: Patches predicted with different PE schemes.

## 3.2 TOKEN STRUCTURE AND DESIGN OF REPRESENTATION $\rho$ FOR NVS

In the previous experiment, tokens were simplified to comprise an entire image feature and an associated camera extrinsic. This differs from typical NVS model token structures where patched image tokens are used, and each of the tokens can be linked not only to a camera transformation but also to a 2D location within an image. To adapt GTA to such NVS models, we now describe how we associate each feature with a geometric attribute and outline one specific design choice for $\rho$.

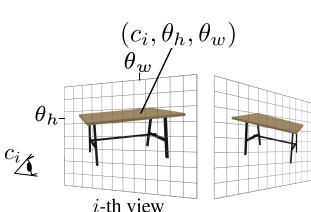

Fig. 4: Geometric attributes.

**Token structure:** We follow a common way to compose the input tokens for the transformer as in (Sajjadi et al., 2022b; Du et al., 2023). We assume that for each view, we have image patches or pixels of the size of $H \times W$, and each patch or pixel token consists of a pair of a feature value $x \in \mathbb{R}^d$ and geometric attributes that are a camera extrinsic $c \in SE(3)$ and a 2D image position. For image PE, it would be natural to encode each position as an element of the 2D translation group $T(2)$. However, we found, similarly to the Fourier feature embeddings used in APE and RPE and rotary PE (Su et al., 2021), encoding the image positions as elements of the 2D rotation group $SO(2)$ exhibits better performance than using $T(2)$. Thus, we represent each image position as an element of the direct product of the two $SO(2)$ groups: $(\theta_h, \theta_w) \in SO(2) \times SO(2)$ where $\theta_h, \theta_w \in [0, 2\pi]$. Here, we identify the $SO(2)$ element with the 2D rotation angle. We associate the top left patch (or pixel) with the value $(0, 0)$, while the bottom right patch corresponds to $(2\pi(H-1)/H, 2\pi(W-1)/W)$. For the intermediate patches, we compute their values using linear interpolation of the angle values between the top left and bottom right patches. Overall, we represent the geometric attribute of each token of the $i$-th view by

$$g := (c_i, \theta_h, \theta_w) \in SE(3) \times SO(2) \times SO(2) =: G. \qquad (8)$$

Fig. 4 illustrates how we represent each geometric attribute of each token.

**Design of $\rho$:** What representation to use is a design choice similar to the design choice of the embedding in APE and RPE. As a specific design choice for the representation for NVS tasks, we propose to compose $\rho_g$ by the direct sum of multiple irreducible representation matrices, each responding to a specific component of the group $G$. Specifically, $\rho_g$ is composed of four different types of representations and is expressed in block-diagonal form as follows:

$$\rho_g := \sigma_{\text{cam}}^{\oplus s}(c) \oplus \sigma_{\text{rot}}^{\oplus t}(r) \oplus \sigma_h^{\oplus u}(\theta_h) \oplus \sigma_w^{\oplus v}(\theta_w), \qquad (9)$$

where "$\oplus$" denotes block-concatenation $A \oplus B = \left[\begin{smallmatrix} A & 0 \\ 0 & B \end{smallmatrix}\right]$ and $A^{\oplus a}$ indicates repeating the block concatenation of $A$ a total of $a$ times. We introduce an additional representation $\sigma_{\text{rot}}(r)$ that captures

Table 1: **Components of $\rho_g$.**

| | $\sigma_{\text{cam}}(c)$ | $\sigma_{\text{rot}}(r)$ | $\sigma_h(\theta_h)$ | $\sigma_w(\theta_w)$ |
|---|---|---|---|---|
| matrix form | $\begin{bmatrix} R & T \\ 0 & 1 \end{bmatrix}$ | $\begin{bmatrix} D_r^{(l_1)} & & \\ & \ddots & \\ & & D_r^{(l_{N_{\text{rot}}})} \end{bmatrix}$ | $\begin{bmatrix} M_{\theta_h}^{(f_1)} & & \\ & \ddots & \\ & & M_{\theta_h}^{(f_{N_h})} \end{bmatrix}$ | $\begin{bmatrix} M_{\theta_w}^{(f_1)} & & \\ & \ddots & \\ & & M_{\theta_w}^{(f_{N_w})} \end{bmatrix}$ |
| multiplicity | $s$ | $t$ | $u$ | $v$ |

Table 2: **Test metrics.** Left: CLEVR-TR, Right: MSN-Hard. †Models are trained and evaluated on MultiShapeNet, not MSN-Hard. They are different but generated from the same distribution.

| | PSNR↑ | | PSNR↑ | LPIPS ↓ | SSIM↑ |
|---|---|---|---|---|---|
| APE | 33.66 | LFN† (Sitzmann et al., 2021) | 14.77 | 0.582 | 0.328 |
| RPE | 36.08 | PixelNeRF† (Yu et al., 2021b) | 21.97 | 0.332 | 0.689 |
| SRT | 33.51 | SRT (Sajjadi et al., 2022b) | 24.27 | 0.368 | 0.741 |
| RePAST | 37.27 | RePAST (Safin et al., 2023) | 24.48 | 0.348 | 0.751 |
| GTA (Ours) | **39.63** | SRT+GTA (Ours) | **25.72** | **0.289** | **0.798** |

Context images

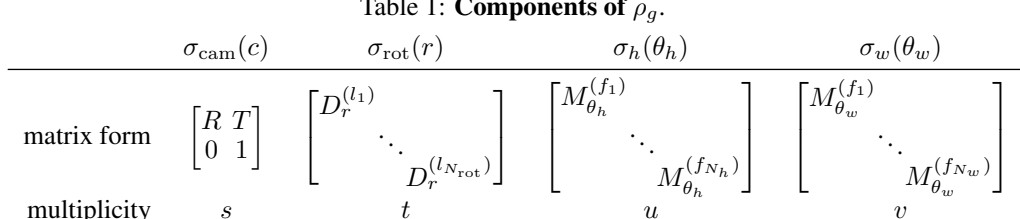

SRT    RePAST    GTA (Ours)    Ground truth

Fig. 5: **Qualitative results on MSN-Hard**.

Fig. 6: **Validation PSNR curves on MSN-Hard.**

only the rotational information of $c$, with which we find moderate improvements in performance. Table 1 summarizes the matrix form we use for each representation. Specifically, $M_\theta^{(f)}$ is a 2D rotation matrix with frequency $f$ that is analogous to the frequency parameter used in Fourier feature embeddings in APE and RPE. $D_r^{(l)}$ can be thought of as the 3D version of $M_\theta^{(f)}$. Please refer to Appendix A.2 for more detailed descriptions of these matrices. Fig. 9 in the Appendix displays the actual representation matrices used in our experiments. The use of the Kronecker product is also a typical way to compose representations, which we describe in Appendix A.3.2.

## 4 EXPERIMENTAL EVALUATION

We conducted experiments on several sparse NVS tasks to evaluate GTA and compare the reconstruction quality with different PE schemes as well as existing NVS methods.

**Datasets:** We evaluate our method on two synthetic 360° datasets with sparse and wide baseline views (*CLEVR-TR* and *MSN-Hard*) and on two datasets of real scenes with distant views (*RealEstate10k* and *ACID*). We train a separate model for each dataset and describe the properties of each dataset below. CLEVR with translation and rotation (CLEVR-TR) is a multi-view version of CLEVR (Johnson et al., 2017) that we propose. It features scenes with randomly arranged basic objects captured by cameras with azimuth, elevation, and translation transformations. We use this

Table 3: **Results on RealEstate10k and ACID**. Top: NeRF methods. Bottom: transformer methods.

| | RealEstate10k | | | ACID | | |
| | PSNR↑ | LPIPS↓ | SSIM↑ | PSNR↑ | LPIPS↓ | SSIM↑ |
|---|---|---|---|---|---|---|
| PixelNeRF (Yu et al., 2021b) | 13.91 | 0.591 | 0.460 | 16.48 | 0.628 | 0.464 |
| StereoNeRF (Chibane et al., 2021) | 15.40 | 0.604 | 0.486 | – | – | – |
| IBRNet (Wang et al., 2021a) | 15.99 | 0.532 | 0.484 | 19.24 | 0.385 | 0.513 |
| GeoNeRF (Johari et al., 2022) | 16.65 | 0.541 | 0.511 | – | – | – |
| MatchNeRF (Chen et al., 2023) | **23.06** | 0.258 | 0.830 | – | – | – |
| GPNR (Suhail et al., 2022) | 18.55 | 0.459 | 0.748 | 17.57 | 0.558 | 0.719 |
| Du et al. (2023) | 21.65 | 0.285 | 0.822 | 23.35 | 0.334 | 0.801 |
| Du et al. (2023) + GTA (Ours) | 22.85 | **0.255** | **0.850** | **24.10** | **0.291** | **0.824** |

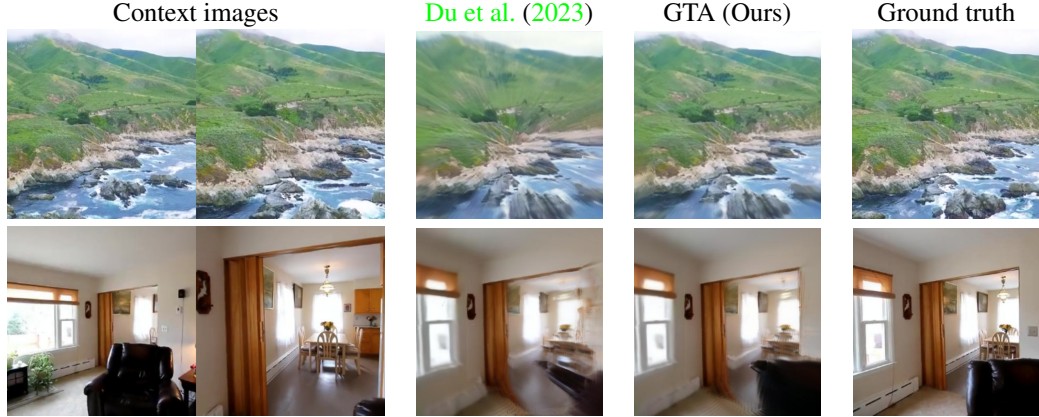

| Context images | Du et al. (2023) | GTA (Ours) | Ground truth |

Fig. 7: **Qualitative results.** Top: ACID, Bottom: RealEstate10k.

dataset to measure the ability of models to understand the underlying geometry of scenes. We set the number of context views to 2 for this dataset. Generating 360° images from 2 context views is challenging because parts of the scene will be unobserved. The task is solvable because all rendered objects have simple shapes and textures. This allows models to infer unobserved regions if they have a good understanding of the scene geometry. MultiShapeNet-Hard (MSN-Hard) is a challenging dataset introduced in Sajjadi et al. (2022a;b). Up to 32 objects appear in each scene and are drawn from 51K ShapeNet objects (Chang et al., 2015), each of which can have intricate textures and shapes. Each view is captured from a camera pose randomly sampled from 360° viewpoints. Objects in test scenes are withheld during training. MSN-Hard assesses both the understanding of complex scene geometry and the capability to learn strong 3D object priors. Each scene has 10 views, and following Sajjadi et al. (2022a;b), we use 5 views as context views and the remaining views as target views. RealEstate10k (Zhou et al., 2018) consists of real indoor and outdoor scenes with estimated camera parameters. ACID (Liu et al., 2021) is similar to RealEstate10k, but solely includes outdoor scenes. Following Du et al. (2023), during training, we randomly select two context views and one intermediate target view per scene. At test time, we sample distant context views with 128 time-step intervals and evaluate the reconstruction quality of intermediate views.

**Baselines:** Scene representation transformer (SRT) (Sajjadi et al., 2022b), a transformer-based NVS method, serves as our baseline model on CLEVR-TR and MSN-Hard. SRT is a similar architecture to the one we describe in Fig. 2, but instead of the extrinsic matrices, SRT encodes the ray information into the architecture by concatenating Fourier feature embeddings of rays to the input pixels of the encoder. SRT is an APE-based model. Details of the SRT rendering process are provided in Appendix C.2.1 and Fig. 15. We also train another more recent transformer-based NVS model called RePAST (Safin et al., 2023). This model is a variant of SRT and encodes ray information via an RPE scheme, where, in each attention layer, the ray embeddings are added to the query and key vectors. The rays linked to the queries and keys are transformed with the extrinsic matrix associated with a key-value token pair, before feeding them into the Fourier embedding functions, to represent both rays in the same coordinate system. RePAST is the current state-of-the-art method

Table 4: **PE schemes.** MLN: Modulated layer normalization (Hong et al., 2023; Liu et al., 2023a). ElemMul: Element-wise Multiplication. GBT: geometry-biased transformers (Venkat et al., 2023). FM: Frustum Embedding (Liu et al., 2022). RoPE+FTL: (Su et al., 2021; Worrall et al., 2017).

| | MLN | SRT | ElemMul | GBT | FM | RoPE+FTL | GTA |
|---|---|---|---|---|---|---|---|
| PSNR↑ | 32.48 | 33.21 | 34.74 | 35.63 | 37.23 | 38.18 | **38.99** |

Table 5: **Effect of the transformation on** $V$. Left: Test PSNRs on ClEVR-TR and MSN-Hard. Right: Inception scores (IS) and FIDs of DiT-B/2 (Peebles & Xie, 2023) on 256x256 ImageNet.

| | CLEVR-TR | MSN-Hard |
|---|---|---|
| No $\rho_g$ on $V$ | 36.54 | 23.77 |
| GTA | **38.99** | **24.58** |

| | IS↑ | FID-50K↓ |
|---|---|---|
| DiT (Peebles & Xie, 2023) | 145.3 | 7.02 |
| DiT + 2D-RoPE | 151.8 | 6.26 |
| DiT + GTA | **158.2** | **5.87** |

on MSN-Hard. The key difference between GTA and RePAST is that the relative transformation is applied directly to QKV features in GTA, while it is applied to rays in RePAST.

For RealEstate10k and ACID, we use the model proposed in Du et al. (2023), which is the state-of-the-art model on those datasets, as our baseline. Their model is similar to SRT, but has architectural improvements and uses an epipolar-based token sampling strategy. The model encodes extrinsic matrices and 2D image positions to the encoder via APE, and also encodes rays associated with each query and context image patch token in the decoder via APE.

We implement our model by extending those baselines. Specifically, we replace all attention layers in both the encoder and decoder with GTA and remove any vector embeddings of rays, extrinsic matrices, and image positions from the model. We train our models and baselines with the same settings within each dataset. We train each model for 2M and 4M iterations on CLEVR-TR and MSH-Hard and for 300K iterations on both RealEstate10k and ACID, respectively. We report the reproduced numbers of baseline models in the main tables and show comparisons between the reported values and our reproduced results in Table 16 and Table 17 in Appendix C.2. Please also see Appendix C.2 for more details about our experimental settings.

**Results:** Tables 2 and 3 show that GTA improves the baselines in all reconstruction metrics on all datasets. Fig. 5 shows that on MSN-Hard, the GTA-based model renders sharper images with more accurate reconstruction of object structures than the baselines. Fig. 7 shows that our GTA-based transformer further improves the geometric understanding of the scenes over Du et al. (2023) as evidenced by the sharper results and the better recovered geometric structures. Appendix D provides additional qualitative results. Videos are provided in the supplemental material. We also train models, encoding 2D positions and camera extrinsics via APE and RPE for comparison. See Appendix C.2.3 for details. Fig. 6 shows that GTA-based models improve learning efficiency over SRT and RePAST by a significant margin and reach the same performance as RePAST using only 1/6 of the training steps on MSN-Hard. GTA also outperforms RePAST in terms of wall-clock time as each gradient update step is slightly faster than RePAST, see also Table 14 in Appendix B.8.

**Comparison to other PE methods:** We compare GTA with other PE methods on CLEVR-TR. All models are trained for 1M iterations. See Appendix C.2.4 for the implementation details. Table 4 shows that GTA outperforms other PE schemes. GTA is better than RoPE+FTL, which uses RoPE (Su et al., 2021) for the encoder-decoder transformer and transforms latent features of the encoder with camera extrinsics (Worrall et al., 2017). This shows the efficacy of the layer-wise geometry-aware interactions in GTA.

**Effect of the transformation on** $V$**:** Rotary positional encoding (RoPE) (Su et al., 2021; Sun et al., 2022) is similar to the $SO(2)$ representations in GTA. An interesting difference to GTA is that the RoPE only applies transformations to query and key vectors, and not to the value vectors. In our setting, this exclusion leads to a discrepancy between the coordinate system of the key and value vectors, both of which interact with the tokens from which the query vectors are derived. Table 5 left shows that removing the transformation on the value vectors leads to a significant drop in performance in our NVS tasks. Additionally, Table 5 right shows the performance on the ImageNet

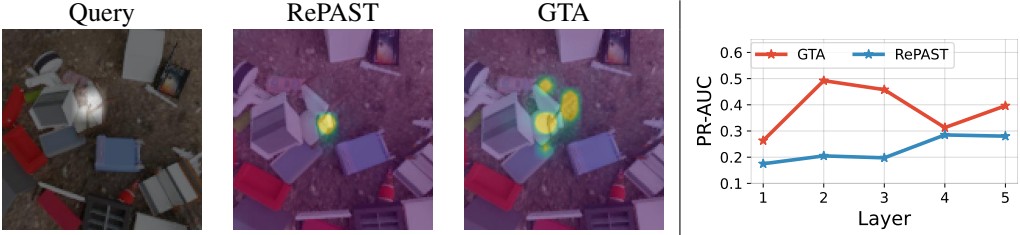

Fig. 8: **Attention analysis.** Given a query token (white region), the attention weights on a context view are visualized. GTA can identify the shape of the object that corresponds to the given query. Right: Quantitative evaluation of alignments between attention matrices and object masks.

Table 6: **Representation design.** Test PSNRs of models trained for 1M iterations.

(a) (✓): representations are not used in the encoder. Camera and image position are needed in the decoder to identify which pixel to render.

| $SE(3)$ | $SO(2)$ | $SO(3)$ | CLEVR-TR | MSN-Hard |
|---|---|---|---|---|
| ✓ | (✓) | | 37.45 | 20.33 |
| (✓) | ✓ | | 38.26 | 23.82 |
| ✓ | ✓ | | 38.99 | 24.58 |
| ✓ | ✓ | ✓ | **39.00** | **24.80** |

(b) Image positions encodings. *Single frequency.

| | CLEVR-TR | MSN-Hard |
|---|---|---|
| $SE(3) + T(2)$ | 37.20 | 23.69 |
| $SE(3) + SO(2)^*$ | 38.82 | 23.98 |
| $SE(3) + SO(2)$ | **38.99** | **24.58** |

generative modeling task with diffusion models. Even on this purely 2D task, the GTA mechanism is better compared to RoPE as an image positional encoding method (For more details of the diffusion experiment, please refer to Appendix C.3).

**Object localization:** As demonstrated in Fig. 8 on MSN-Hard, the GTA-based transformer not only correctly finds patch-to-patch associations but also recovers *patch-to-object* associations already in the second attention layer of the encoder. For quantitative evaluation, we compute precision-recall-AUC (PR-AUC) scores based on object masks provided by MSN-Hard. In short, the score represents, given a query token belonging to a certain object instance, how well the attention matrix aligns with the object masks across all context views. Details on how we compute PR-AUC are provided in Appendix B.7. The PR-AUCs for the second attention layer are 0.492 and 0.204 with GTA and RePAST, respectively, which shows that our GTA-based transformer quickly identifies where to focus attention at the object level.

**Representation design:** Table 6a shows that, without camera encoding ($SE(3)$) or image PE ($SO(2)$) in the encoder, the reconstruction quality degrades, showing that both representations are helpful in aggregating multi-view features. Using $SO(3)$ representations causes a moderate improvement on MSN-Hard and no improvement on CLEVR-TR. A reason for this could be that MSN-Hard consists of a wide variety of objects. By using the $SO(3)$ representation, which is invariant to camera translations, the model may be able to encode object-centric features more efficiently. Table 6b confirms that similar to Fourier feature embeddings used in APE and RPE, multiple frequencies of the $SO(2)$ representations benefit the reconstruction quality.

## 5 CONCLUSION

We have proposed a novel geometry-aware attention mechanism for transformers and demonstrated its efficacy by applying it to sparse wide-baseline novel view synthesis tasks. A limitation of GTA is that GTA and general PE schemes rely on known poses or poses estimated by other algorithms, such as COLMAP (Schönberger & Frahm, 2016). An interesting future direction is to simultaneously learn the geometric information together with the forward propagation of features in the transformer. Developing an algorithm capable of autonomously acquiring such structural information solely from observations, specifically seeking a *universal learner* for diverse forms of structure akin to human capacity, represents a captivating avenue for future research.

## 6 ACKNOWLEDGEMENT

Takeru Miyato, Bernhard Jaeger, and Andreas Geiger were supported by the ERC Starting Grant LEGO-3D (850533) and the DFG EXC number 2064/1 - project number 390727645. The authors thank the International Max Planck Research School for Intelligent Systems (IMPRS-IS) for supporting Bernhard Jaeger. We thank Mehdi Sajjadi and Yilun Du for their comments and guidance on how to reproduce the results and thank Karl Stelzner for his open-source contribution of the SRT models. We thank Haofei Xu and Anpei Chen for conducting the MatchNeRF experiments. We also thank Haoyu He, Gege Gao, Masanori Koyama, Kashyap Chitta, and Naama Pearl for their feedback and comments. Takeru Miyato acknowledges his affiliation with the ELLIS (European Laboratory for Learning and Intelligent Systems) PhD program.

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

# Appendix

## Table of Contents

## A   ADDITIONAL DETAILS OF GTA

Algorithm 1 provides an algorithmic description based on Eq. (6) for single-head self-attention. For multi-head attention, we simply apply the group representations to all $QKV$ vectors of each head.

### A.1   COMPUTATIONAL COMPLEXITY

Since the $\mathbf{P_g} \odot \cdot$ operation is an $n$-times multiplication of a $d \times d$ matrix with a $d$-dimensional vector, the computational complexity of additional computation for our attention over the vanilla attention is $O(nd^2)$. This can be reduced by constructing the representation matrix with a block diagonal, with each block being small. If we keep the largest block size of the representation constant against $n$ and $d$, then the order of the $\mathbf{P_g} \odot \cdot$ operation becomes $O(nd)$. Thus, if $d_{\max}$ is relatively small, or if we increase $n$ or $d$, the computation overhead of the $\odot$ operation becomes negligible compared to the computation times of the other components of a transformer, which are $O(n^2 d)$ for attention and $O(nd^2)$ for feedforward layers. In our experiments, we use a block-structured representation with a maximum block size of 5 (see  Section 3.2 and  Fig. 9).

### A.2   DETAILS OF THE REPRESENTATION MATRICES

$\rho_g$ is composed of four different types of representations $\rho_c, \rho_r, \rho_{\theta_h}, \rho_{\theta_w}$ with the multiplicities of $s, t, u, v$. Below, we describe the details of each representation.

---

**Algorithm 1** GTA for single head self-attention.

---

Input: Input tokens: $X \in \mathbb{R}^{N \times d}$, group representations: $\mathbf{P_g} = [\rho_{g_1}, \rho_{g_2}, ..., \rho_{g_N}]$, and weights: $W^Q, W^K, W^V \in \mathbb{R}^{d \times d}$.

1. Compute query, key, and value from $X$:

$$Q = XW^Q, K = XW^K, V = XW^V.$$

2. Transform each variable with the group representations:

$$Q \leftarrow \mathbf{P_g}^{\mathrm{T}} \circledcirc Q, K \leftarrow \mathbf{P_g}^{-1} \circledcirc K, V \leftarrow \mathbf{P_g}^{-1} \circledcirc V$$

3. Compute $QKV$ attention in the same way as in the vanilla attention:

$$O = \mathrm{softmax}\left(\frac{QK^{\mathrm{T}}}{\sqrt{d}}\right) V$$

4. Apply group representations to $O$:

$$O \leftarrow \mathbf{P_g} \circledcirc O$$

5. Return $O$

---

$\sigma_{\mathrm{cam}}(c)$: We use a homogenous rigid transformation as the representation of $c \in SE(3)$:

$$\sigma_{\mathrm{cam}}(c) := \begin{bmatrix} R & T \\ 0 & 1 \end{bmatrix} \in \mathbb{R}^{4 \times 4}. \tag{10}$$

$\sigma_{\mathrm{rot}}(r)$: We compose $\sigma_{\mathrm{rot}}(r)$ via block concatenation of Wigner-D-matrices (Chirikjian, 2000).

$$\sigma_{\mathrm{rot}}(r) := \bigoplus_k \sigma_{\mathrm{rot}}^{(l_k)}(r), \ \sigma_{\mathrm{rot}}^{(l)}(r) := D_r^{(l)} \in \mathbb{R}^{(2l+1) \times (2l+1)} \tag{11}$$

where $D_r^{(l)}$ is $l$-th Wigner-D-matrix given $r$. Here, $\bigoplus_{a \in S} A^{(a)} := A^{(a_1)} \oplus \cdots \oplus A^{(a_{|S|})}$ and we omit the index set symbol from the above equation. We use these matrices because Wigner-D-matrices are the *only irreducible representations* of $SO(3)$. Any linear representation $\sigma_r, r \in SO(3)$ is equivalent to a direct sum of the matrices under a similarity transformation (Chirikjian, 2000).

$\sigma_h(\theta_h)$ **and** $\sigma_w(\theta_w)$: Similar to $\sigma_{\mathrm{rot}}(r)$, we use 2D rotation matrices with different frequencies for $\sigma_h(\theta_h)$ and $\sigma_w(\theta_w)$. Specifically, for $\sigma_h(\theta_h)$ given a set of frequencies $\{f_k\}_{k=1}^{N_h}$, we define the representation as follows:

$$\sigma_h(\theta_h) := \bigoplus_k \sigma_h^{(f_k)}(\theta_h), \ \sigma_h^{(f)}(\theta_h) := M_{\theta_h}^{(f)} = \begin{bmatrix} \cos(f\theta_h) & -\sin(f\theta_h) \\ \sin(f\theta_h) & \cos(f\theta_h) \end{bmatrix} \in \mathbb{R}^{2 \times 2}. \tag{12}$$

$\sigma_w(\theta_w)$ is defined analogously.

We use the following strategy to choose the multiplicities $s, t, u, v$ and frequencies $\{l\}, \{f_h\}, \{f_w\}$:

1. Given the feature dimension $d$ of the attention layer, we split the dimensions into three components based on the ratio of $2 : 1 : 1$.
2. • We apply $\sigma_{\mathrm{cam}}^{\oplus s}$ to the first half of the dimensions. As $\sigma_{\mathrm{cam}}$ does not possess multiple frequencies, its multiplicity is set to $d/8$.
   • $\sigma_{\mathrm{rot}}^{\oplus r}$ is applied to a quarter of the dimensions. For the frequency parameters $\{l\}$ of $\sigma_{\mathrm{rot}}$, we consistently use the 1st and 2nd degrees of the Winger-D matrices. Considering the combined sizes of these matrices is 8, the multiplicity for $\sigma_{\mathrm{rot}}$ becomes $d/32$.
   • For the remaining $1/4$ of the dimensions of each QKV vector, we apply both $\sigma_h^{\oplus t}$ and $\sigma_w^{\oplus u}$. Regarding the frequency parameters $\{f_h\}, \{f_w\}$, we utilize $d/16$ octaves with the maximum frequency set at 1 for both $\sigma_h$ and $\sigma_w$. The multiplicities for these are both 1.

Based on this strategy, we use the multiplicities and frequencies shown in Table 7. Also Fig. 9 displays the actual representation matrices used on the MSN-Hard dataset.

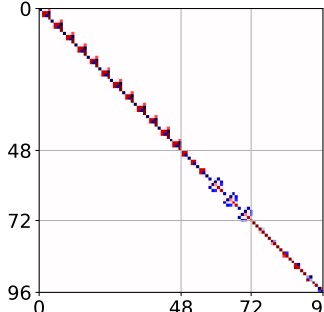 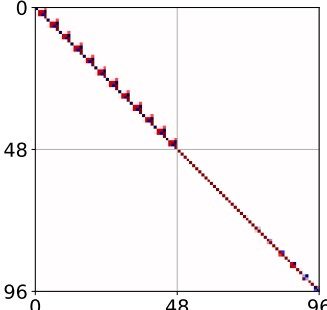

Fig. 9: **Representation matrices on MSN-Hard.** Left: with $SO(3)$, Right: without $SO(3)$. Left: Dimensions 1-48 correspond to $\sigma_{\text{cam}}^{\oplus 12}(c)$, dimensions 49-72 correspond to $\sigma_{\text{rot}}^{\oplus 3}(r)$, and dimensions 73-96 correspond to $\sigma_h(\theta_h)$ and $\sigma_w(\theta_w)$. Right: Dimensions 1-48 correspond to $\sigma_{\text{cam}}^{\oplus 12}(c)$ and dimensions 49-96 correspond to $\sigma_h(\theta_h)$ and $\sigma_w(\theta_w)$.

Table 7: **Multiplicity and frequency parameters.** Here, $d$ is the dimensions of each attention head. Since the baseline model on RealEsate10k and ACID uses different feature sizes for query-key vectors and value vectors, we also use different sizes of representation matrices for each feature.

|  | $d$ | $\{s, t, u, v\}$ | $\{l_1, ..., l_{N_{\text{rot}}}\}$ | $\{f_1, ..., l_{N_{\{\text{h,w}\}}}\}$ |
|---|---|---|---|---|
| CLEVR-TR | 64 | $\{8, 3, 1, 1\}$ | $\{1, 2\}$ | $\{1, ..., 1/2^3\}$ |
| CLEVR-TR wo/ $SO(3)$ |  | $\{8, 0, 1, 1\}$ | $-$ | $\{1, 1/2, 1/4, ..., 1/2^7\}$ |
| MSH-Hard | 96 | $\{12, 3, 1, 1\}$ | $\{1, 2\}$ | $\{1, 1/2, 1/4, ..., 1/2^5\}$ |
| MSH-Hard wo/ $SO(3)$ |  | $\{12, 0, 1, 1\}$ | $-$ | $\{1, 1/2, 1/4, ..., 1/2^{11}\}$ |
| Realestate10k and ACID (Encoder) | 64 | $\{8, 3, 1, 1\}$ | $\{1, 2\}$ | $\{1, ..., 1/2^3\}$ |
| Realestate10k and ACID (Decoder, key) | 128 | $\{16, 6, 1, 1\}$ | $\{1, 2\}$ | $\{1, ..., 1/2^7\}$ |
| Realestate10k and ACID (Decoder, value) | 256 | $\{32, 12, 1, 1\}$ | $\{1, 2\}$ | $\{1, ..., 1/2^{15}\}$ |

## A.3 VARIANTS OF GTA

Here we would like to introduce two variants of GTA. The one is the Euclidean version of GTA where we use the Euclidean distance for the attention similarity. The other one is GTA with a group representation composed of the Kronekcer product of smaller representations. We see in Table 8 that the performances of those variants of GTA are a little degraded but relatively close to the original GTA. We will detail each variant in the following sections.

### A.3.1 EUCLIDEAN GTA

The unconventional aspect of Eq. (6) is the presence of the transpose in the transformation of the query vectors. The transpose is necessary for having the reference coordinate invariance, and the need arises from the fact that the dot-product similarity is not invariant under $SE(3)$ transformations when the translation component is non-zero. To ensure both reference coordinate invariance and consistent transformations across the $Q, K, V$ vectors, one can utilize the Euclidean similarity for computing the attention matrix instead of the dot-product similarity. The formula for the self-attention layer with squared Euclidean distance is given by:

$$O := \text{Attn}_{\text{Euclid}}(Q, K, V) = \text{softmax}(\mathcal{E}(Q, K))V, \tag{13}$$

$$\text{where } \mathcal{E}(Q, K) \in \mathbb{R}^{N \times N}, \mathcal{E}_{ij}(Q, K) = -\|Q_i - K_j\|_2^2. \tag{14}$$

Then the Euclidean version of GTA (GTA-Euclid) is written in the following form:

$$O = \mathbf{P_g} \odot \text{Attn}_{\text{Euclid}}\left(\mathbf{P_g}^{-1} \odot Q, \mathbf{P_g}^{-1} \odot K, \mathbf{P_g}^{-1} \odot V\right). \tag{15}$$

Eq. (15) possesses the reference coordinate invariance property since the square distance is preserved under rigid transformations. The numbers of Table 8 are produced under the same setting as the original GTA, except that we replaced the dot-product attention with the Euclidean attention that we describe above.

### A.3.2 KRONECKER GTA

Another typical way to compose a representation matrix is using the Kronecker product. The Kronecker product of two square matrices $A, B \in \mathbb{R}^{m \times m}, \mathbb{R}^{n \times n}$ is defined as:

$$A \otimes B = \begin{bmatrix} a_{11}B & \cdots & a_{1m}B \\ \vdots & \ddots & \vdots \\ a_{m1}B & \cdots & a_{mm}B \end{bmatrix} \in \mathbb{R}^{mn \times mn}. \tag{16}$$

The important property of the Kronecker product is that the Kronecker product of two representations is also a representation: $(\rho_1 \otimes \rho_2)(gg') = (\rho_1 \otimes \rho_2)(g)(\rho_1 \otimes \rho_2)(g')$ where $(\rho_1 \otimes \rho_2)(g) := \rho_1(g) \otimes \rho_2(g)$. We implement the Kronecker version of GTA, which we denote GTA-Kronecker, where we use the Kronecker product of the $SE(3)$ representation and $SO(2)$ representations as a representation $\rho_g$:

$$\rho_g = \rho_{\text{cam}}(c) \otimes (\rho_h(\theta_h) \oplus \rho_w(\theta_w)), \text{where } g = (c, \theta_h, \theta_w). \tag{17}$$

In the results presented in Table 8, the multiplicity of $\rho_{\text{cam}}$, $\rho_h$, $\rho_g$ are set to 1, and the number of frequencies for both $\rho_h$ and $\rho_w$ is set to 4 on CLEVR-TR and 6 on MSN-Hard.

Table 8: **Results with different representation forms.**

|  | CLEVR-TR | MSN-Hard |
| --- | --- | --- |
| GTA-Kronecker | 38.32 | 24.52 |
| GTA-Euclid | 38.59 | 24.75 |
| GTA | **38.99** | **24.80** |

### A.4 RELATION TO EQUIVARIANT AND GAUGE EQUIVARIANT NETWORKS

The gauge transform used in gauge equivariant networks (Cohen et al., 2019; De Haan et al., 2021; He et al., 2021; Brandstetter et al., 2022) is related to the relative transform $\rho(g_i g_j^{-1})$ used in our attention mechanism. However, the equivariant models and ours differ because they are built on different motivations. In short, the gauge equivariant layers are built to preserve the feature field structure determined by a gauge transformation. In contrast, since image features themselves are not 3D-structured, our model applies the relative transform only on the query and key-value pair in the attention mechanism but does not impose equivariance on the weight matrices of the attention and the feedforward layers. The relative transformation in GTA can be thought of as a form of guidance that helps the model learn structured features within the attention mechanism from the initially unstructured raw multi-view images.

Brehmer et al. (2023) introduce geometric algebra (GA) to construct equivariant transformer networks. Elements of GA are themselves operators that can act on GA, which may enable us to construct expressive equivariant models by forming bilinear layers that allow interactions between different multi-vector subspaces. In NVS tasks, where the input consists of raw images lacking geometric structure, directly employing such equivariant models may not be straightforward. However, integrating GA into the GTA mechanism could potentially enhance the network's expressivity, warranting further investigation.

# B    ADDITIONAL EXPERIMENTAL RESULTS

## B.1    TRAINING CURVES ON CLEVR-TR AND MSN-HARD.

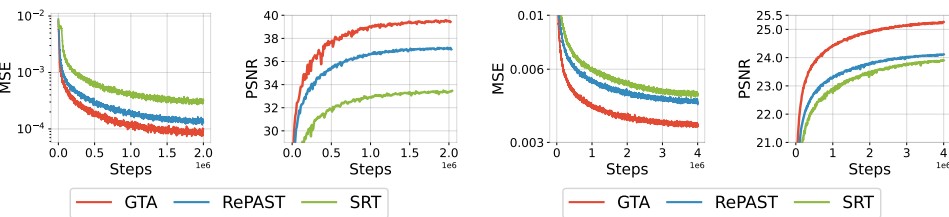

Fig. 10: **Training and validation curves.** Left: CLEVR-TR, Right: MSN-Hard.

## B.2    RESULTS WITH HIGHER RESOLUTION

Table 9 shows results on RealEstate10k with 384x384 resolutions (1.5 times larger height and width than in Table 3). We see that GTA also improves over the baseline model at higher resolution.

Table 9: **384×384 resolution on RealEstate10K**

|  | PSNR↑ | LPIPS↓ | SSIM↑ |
|---|---|---|---|
| Du et al. (2023) | 21.77 | 0.316 | 0.848 |
| Du et al. (2023) + GTA (Ours) | **22.77** | **0.290** | **0.864** |

## B.3    RESULTS WITH 3 CONTEXT VIEWS

We train models with 3-context views and show the results in Table 10. We see that GTA is also better with context views more than 2.

Table 10: **Results with different numbers of context views on RealEstate10K**

|  | 2-view PSNR↑ | 3-view PSNR↑ |
|---|---|---|
| Du et al. (2023) | 21.65 | 21.88 |
| Du et al. (2023) + GTA (Ours) | **22.85** | **23.22** |

## B.4    ROBUSTNESS TO CAMERA NOISE

Table 11 shows results on CLEVR-TR with a presence of camera noise. We train RePAST and GTA with camera noise added to each camera extrinsic of the second view. We perturb camera extrinsics by adding Gaussian noise to the coefficients of the $SE(3)$-Lie algebra basis. The mean and variance of the noise is set to $(m, \sigma) = (0, 0.1)$ during training. GTA shows better performance than RePAST regardless of the noise level.

Table 11: **Test PSNRs with camera noise on CLEVR-TR and MSN-Hard**. $\sigma_{\text{test}}$ indicates the noise strength at test time. [†]No noise injection during training.

|  | CLEVR-TR | | MSN-Hard | |
|---|---|---|---|---|
| $\sigma_{\text{test}}$ | 0.01 | 0.1 | 0.01 | 0.1 |
| RePAST (Safin et al., 2023) | 35.26 | 35.14 | 22.76 | 22.60 |
| SRT+GTA (Ours) | **36.66** | **36.57** | **24.06** | **24.16** |

Table 12: **Test metrics on CLEVR-TR.**

|  | PSNR↑ | LPIPS ↓ | SSIM↑ |
|---|---|---|---|
| APE | 33.66 | 0.161 | 0.960 |
| RPE | 36.06 | 0.159 | 0.971 |
| SRT | 33.51 | 0.158 | 0.960 |
| RePAST | 37.27 | 0.119 | 0.977 |
| GTA (Ours) | **39.63** | **0.108** | **0.984** |

Table 14: **Computational time to perform one gradient step, encode a single scene, and render a single entire image on MSN-Hard (top) and RealEstate10K (bottom)**. All time values are expressed in milliseconds (ms). As for one gradient step time, we only measure time for forward-backward props and weight updates and exclude data loading time. We measure each time on a single A100 with bfloat16 precision for MSN-Hard and float32 precision for RealEstate10K.

| Method | One gradient step | Encoding | Rendering |
|---|---|---|---|
| SRT | 296 | 5.88 | 16.4 |
| RePAST | 394 | 7.24 | 21.4 |
| GTA | 379 | 17.7 | 20.9 |

| Method | One gradient step | Encoding | Rendering |
|---|---|---|---|
| Du et al. (2023) | 619 | 49.8 | $1.42 \times 10^3$ |
| GTA | 806 | 74.3 | $2.05 \times 10^3$ |

## B.5 PERFORMANCE WITH DIFFERENT RANDOM SEEDS

We observe that the performance variance of different random weight initializations is quite small, as shown in Fig. 11, which displays the mean and standard deviation across 4 different seeds. We see that the variance is relatively insignificant compared to the performance difference between the compared methods. Consequently, the results reported above are statistically meaningful.

## B.6 PERFORMANCE DEPENDENCE ON THE REFERENCE COORDINATES

Table 13 highlights the importance of coordinate-choice invariance. "SRT (global coord)" is trained with camera poses that have their origin set to always be in the center of all objects. This setting enables the model to know how far the ray origin is from the center of the scene, therefore enabling the model to easily find the position of the surface of objects that intersect with the ray. We see that SRT's performance heavily depends on the choice of reference coordinate system. Our model is, by construction, invariant to the choice of reference coordinate system of cameras and outperforms even the privileged version of SRT.

Table 13: **Test PSNRs in a setting where global coordinates are shared across scenes.** All numbers show test PSNRs and are produced with models trained for 1M iterations. Note that GTA is invariant to the reference coordinates of the extrinsics, and the performance is not affected by the choice of the reference coordinate system.

| Method | CLEVR-TR | MSH-Hard |
|---|---|---|
| SRT | 32.97 | 23.15 |
| SRT (global coord) | 37.93 | 24.20 |
| GTA wo $SO(3)$ | **38.99** | **24.58** |

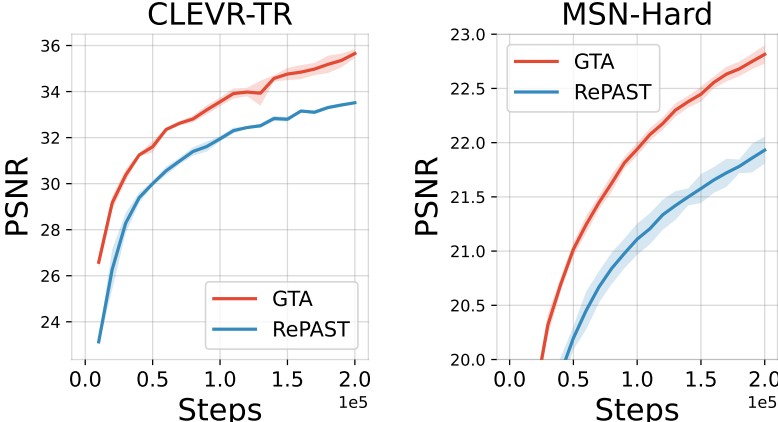

Fig. 11: **Mean and standard deviation plots of validation PSNRs on CLEVR-TR and MSN-Hard**. Due to the heavy computation requirements for training, we only trained models with 200,000 iterations and measured the validation PSNRs during the course of the training.

## B.7 ANALYSIS OF ATTENTION PATTERNS

We conducted an analysis on the attention matrices of the encoders trained on MSN-Hard. We found that the GTA-based model tends to attend to features of different views more than RePAST, which we show in Fig. 12. Furthermore, we see that GTA not only correctly attends to the respective patches of different views, but also can attend to object level regions (Fig. 8 and 13). Surprisingly, these attention patterns are seen at the very beginning of the course of the encoding process: the visualized attention maps are obtained in the 2nd attention layer. To evaluate how well the attention maps $\alpha$ weigh respective object features across views, we compute a retrieval-based metric with instance segmentation masks of objects provided by MSN-Hard. Specifically, given a certain layer's attention maps $\alpha$:

1. We randomly sample the $i$-th query patch token with 2D position $p \in \{1, ..., 16\} \times \{1, ..., 16\}$.
2. We compute the attention map $\bar{\alpha}_i \in [0, 1]^{5*16*16}$ averaged over all heads.
3. We then identify which object belongs to that token's position by looking at the corresponding $8 \times 8$ region of the instance masks. Note that multiple objects can belong to the region.
4. For each belonging object, we compute precision and recall values with $\mathbb{1}[\bar{\alpha}_i > t]$ as prediction and 0–1 masks of the corresponding object as ground truth on all context views, by changing the threshold value $t \in [0, 1]$.
5. In the final step, we calculate a weighted average of the precision and recall values for each object. To determine the weight of each object, we consider the number of pixels assigned to that object's mask within the 8x8 region. We then normalize these weights so that their sum equals to be one.

We collect multiple precision and recall values by randomly sampling scenes and patch positions 2000 times and then compute the average of the collected precision-recall curves. In Fig. 14, we show averaged precision-recall curves. Table 8 shows the area under the precision-recall curves (PR-AUCs) of each layer. We see that the GTA-based model learns well-aligned attention maps with the ground truth object masks for every layer.

## B.8 COMPUTATIONAL TIME

We measure the time to perform one-step gradient descent, as well as encoding and decoding for each method. Table 14 shows that the computational overhead added by the use of GTA is comparable to RePAST on MSN-Hard. In contrast to GTA and RePAST-based models which encode positional information into every layer, SRT and Du et al. (2023) add positional embeddings only to each encoder and decoder input. As a result, the computational time of SRT for one-step gradient

GTA                                          RePAST

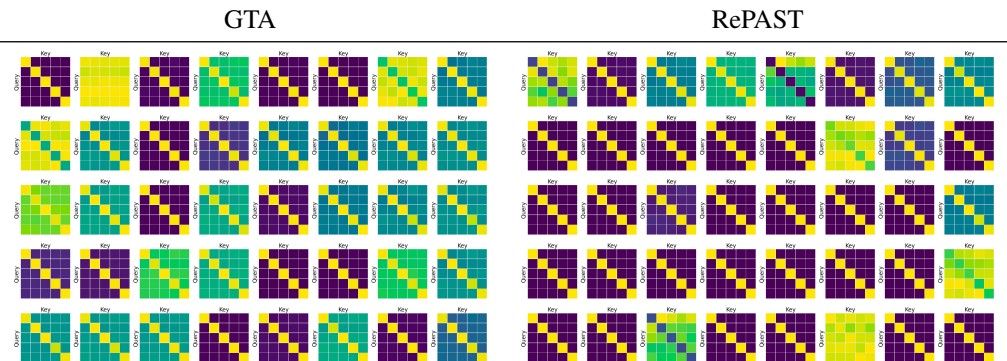

Fig. 12: **Visualization of view-to-view attention maps**. The $(i, j)$-th element of each 5x5 matrix represents the average of attention weights between all pairs of each query token of the $i$-th context view and each key token of the $j$-th context view. The $(l, m)$-th panel shows the weight of the $m$-th head at the $l$-th layer. Yellow and dark purple cells indicate high and low attention weight, respectively. A matrix with high diagonal values means that the corresponding attention head attends *within each view* while with high non-diagonal values means the corresponding attention head attends *across views*.

descent is around 1.3x faster than RePAST and GTA, and that of Du et al. (2023) is 1.3x faster than GTA.

## C    EXPERIMENTAL SETTINGS

### C.1    DETAILS OF THE SYNTHETIC EXPERIMENTS IN SECTION 3.1

We use $10,000$ training and test scenes. For the intrinsics, both the vertical and horizontal sensor width are set to 1.0, and the focal length is set to 4.0, leading to an angle of view of $28°$.

For optimization, we use AdamW (Loshchilov & Hutter, 2017) with weight decay 0.001. For each PE method, we trained multiple models with different learning rates of $\{0.0001, 0.0002, 0.0005\}$ and found 0.0002 to work best for all models, and hence show results with this learning rate. We use three attention layers for both the encoder and the decoder. The image feature dimension is $32 \times 32 \times 3$. This feature is flattened and fed into a 2 layer-MLP to be transformed into the same dimensions as the token dimension $d$. We also apply a 2 layer-MLP to the output of the decoder to obtain the $3,072$ dimensional predicted image feature. The token dimensions $d$ are set to 512 for APE and RPE. As we mention in the descriptions of the synthetic experiment, $\rho_g$ is composed of block concatenation of $3 \times 3$ rotation matrices, and we set $d$ to 510 for GTA, which is divisible by 3. Note that there is no difficulty with the case where $d$ is not divisible by 3. In that case, we can apply $\rho_g$ only to certain components of vectors whose dimensions are divisible by 3 and apply no transformation to the other dimensions. This corresponds to applying a trivial representation, i.e., the identity matrix, to the remaining vectors.

The RPE-based model we designed is a sensible model. For example, if $b^Q = b^K$ and the set of three-dimensional vector blocks of $b^Q$ forms an orthonormal basis, then the inner product of the transformed query and key bias vectors becomes the *trace* of the product of the rotation matrices: $\langle \rho(r)b^Q, \rho(r')b^K \rangle = \text{tr}(R^\text{T}R')$. $\text{tr}(A^\text{T}B)$ is a natural inner product for matrices, by which we can bias the attention weight based on the inner-product-based similarity of matrices. Hence, we initialize each of the biases with vectorized identity matrices.

### C.2    EXPERIMENTAL SETTINGS IN SECTION 4

Table 15 shows dataset properties and hyperparameters that we use in our experiments. We train with 4 RTX 2080 Ti GPUs on CLEVR-TR and with 4 Nvidia A100 GPUs on the other datasets.

GTA

RePAST

GTA

RePAST

GTA

RePAST

Fig. 13: **Additional attention map visualizations.**

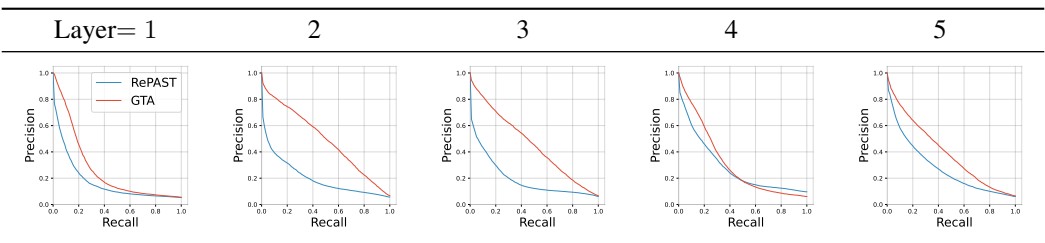

Fig. 14: **Precision-recall curves of the attention matrices of each encoder layer.**

Table 15: **Dataset properties and architecture hyperparameters**. # target pixels indicate how many query pixels are sampled for each scene during training. *We use 12 heads for the attention layers in SRT and 8 heads in RePAST and GTA because 12 head models do not fit into our GPU memory with those methods. †The decoder's attention layers only have a single head. Also, the token dimensions in the decoder are set to 128 for query-key vectors and 256 for value vectors.

| dataset | CLEVR-TR | MSN-Hard | RealEstate10k | ACID |
|---|---|---|---|---|
| # Training scenes | 20,000 | 1,000,000 | 66,837 | 10,974 |
| # Test scenes | 1,000 | 10,000 | 7,192 | 1,910 |
| Batch size | 32 | 64 | 48 | 48 |
| Training steps | 2,000,000 | 4,000,000 | 300,000 | 200,000 |
| Learning rate | 1e-4 | 1e-4 | 5e-4 | |
| # Context views | 2 | 5 | 2 | |
| # Target pixels | 512 | 2,048 | 192 | |
| # Self-attention layers in the encoder | 5 | 5 | 12 | |
| # Cross-attention layers in the decoder | 2 | 2 | 2 | |
| # Heads in attention layers | 6 | 12/8* | $12^{\dagger}$ | |
| Token dimensions | 384 | 768 | $768^{\dagger}$ | |
| MLP dimensions | 768 | 1,536 | 3,072 | |

Table 16: **Performance comparison between numbers reported in Safin et al. (2023) and our reproduced numbers**. Note that Safin et al. (2023) uses 4x larger batch size than available in our experimental setting (4 A100s). The number of iterations for which we train each model is the same as Safin et al. (2023).

| | PSNR↑ | LPIPS$_{VGG/Alex}$ ↓ | SSIM↑ |
|---|---|---|---|
| SRT (Sajjadi et al., 2022b) | 24.56 | NA/0.223 | 0.784 |
| RePAST (Safin et al., 2023) | 24.89 | NA/0.202 | 0.794 |
| SRT | 24.27 | 0.368/0.279 | 0.741 |
| RePAST | 24.48 | 0.348/0.243 | 0.751 |
| SRT+GTA (Ours) | **25.72** | **0.289/0.185** | **0.798** |

**CLEVR-TR and MSN-Hard** CLEVR-TR is synthesized by using Kubric (Greff et al., 2022). The resolution of each image is $240 \times 320$. The camera poses of the dataset include translation, azimuth, and elevation transformations. The camera does not always look at the center of the scene.

MSN-Hard is also a synthetically generated dataset. Up to 32 objects sampled from ShapeNet (Chang et al., 2015) appear in each scene. All 51K ShapeNet objects are used for this dataset, and the training and test sets do not share the same objects with each other. MSN-Hard includes instance masks for each object in a scene, which we use to compute the attention matrix alignment score described in Section 4 and Appendix B.7. The resolution of each image is $128 \times 128$.

We basically follow the same architecture and hyperparameters of the improved version of SRT described in the appendix of Sajjadi et al. (2022a), except that we use AdamW (Loshchilov & Hutter, 2017) with the weight decay set to the default parameter and dropout with a ratio of 0.01 at every attention output and hidden layers of feedforward MLPs.

Since there is no official code or released models available for SRT and RePAST, we train both baselines ourselves and obtain almost comparable but slightly worse results (Table 16). This is because we train the models with a smaller batch size and target ray samples than in the original setting due to our limited computational resources (4 A100s). Note that our model, which is also trained with a smaller batch size, still outperforms the original SRT and RePAST models' scores.

**RealEstate10k and ACID** Both datasets are sampled from videos available on YouTube. At the time we conducted our experiments, some of the scenes used in Du et al. (2023) were no longer available on YouTube. We used scenes $66,837$ and $10,974$ training scenes and $7,192$ and $1,910$ test scenes for RealEstate10k and ACID, respectively. The resolution of each image in the original

Table 17: **Comparison between results reported in Du et al. (2023) (Top) and our reproduced results (Bottom).**

| | RealEstate10k | | | ACID | | |
| --- | --- | --- | --- | --- | --- | --- |
| | PSNR↑ | LPIPS↓ | SSIM↑ | PSNR↑ | LPIPS↓ | SSIM↑ |
| Du et al. (2023) | 21.38 | 0.262 | 0.839 | 23.63 | 0.364 | 0.781 |
| Du et al. (2023) | 21.65 | 0.284 | 0.822 | 23.35 | 0.334 | 0.801 |
| Du et al. (2023) + GTA (Ours) | **22.85** | **0.255** | **0.850** | **24.10** | **0.291** | **0.824** |

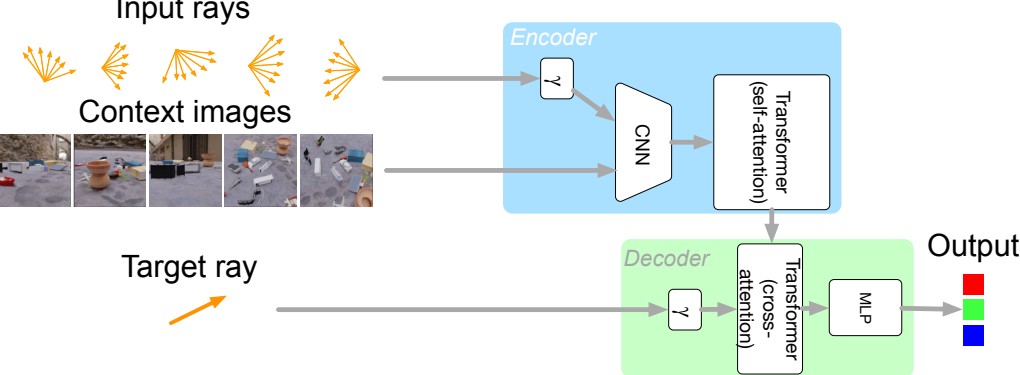

Fig. 15: **Scene representation transformer (SRT) rendering process**. The encoder $E$ consisting of a stack of convolution layers followed by a transformer encoder translates context images into a set representation $S$. The decoder $D$ predicts an RGB pixel value given a target ray and $S$. In our model, every attention layer in both the encoder and decoder is replaced with GTA. We also remove the input and target ray embeddings from the input of the encoder and decoder, respectively. We input a learned constant vector to the decoder instead of the target ray embeddings.

sequences is $360 \times 640$. For training, we apply downsampling followed by a random crop and random horizontal flipping to each image, and the resulting resolution is $256 \times 256$. For test time, we apply downsampling followed by a center crop to each image. The resolution of each processed image is also $256 \times 256$. We follow the same architecture and optimizer hyperparameters of Du et al. (2023). Although the authors of Du et al. (2023) released the training code and their model on RealEstate10k, we observed that the model produces worse results than those reported in their work. The results were still subpar even when we trained models with their code. As a result, we decided to train each model with more iterations (300K) compared to the 100K iterations mentioned in their paper and achieved comparable scores on both datasets. Consequently, we also trained GTA-based models for 300K iterations as well.

### C.2.1 SCENE REPRESENTATION TRANSFORMER (SRT)

**Encoding views:** Let us denote $N_{\text{context}}$-triplets of input view images and their associated camera information by $\mathbf{I} := \{(I_i, c_i, M_i)\}_{i=1}^{N_{\text{context}}}$, where $N_{\text{context}}$ is the number of context views, $I_i \in R^{H \times W \times 3}$ is the $i$-th input RGB image, and $c_i \in \mathbb{R}^{4 \times 4}, M_i \in \mathbb{R}^{3 \times 3}$ are a camera extrinsic and a camera intrinsic matrix associated of the $i$-th view. The SRT encoder $E$ encodes the context of views into scene representation $S$ and is composed of a CNN and a transformer $E_{\text{transformer}}$. First, a 6-layer CNN $E_{\text{CNN}}$ is applied to a ray-concatenated image $I'$ of each view to obtain $(H/D) \times (W/D)$-resolution features:

$$F_i = E_{\text{CNN}}(I'_i) \in \mathbb{R}^{(H/D) \times (W/D) \times d}, \; I'_{ihw} = I_{ihw} \oplus \gamma(r_{ihw}) \qquad (18)$$

where $d$ is the output channel size of the CNN, and $D$ is the downsampling factor, which is set to 8. $\gamma$ is a Fourier embedding function that transforms ray $r = (o, d) \in \mathbb{R}^3 \times \mathcal{S}$ into a concatenation of the Fourier features with multiple frequencies. Each ray $r_{ihw}$ is computed from given camera's extrinsic and intrinsic parameters $(c_i, M_i)$. Here, "$\oplus$" denotes vector concatenation.

Next, a transformer-based encoder $E_{\text{transformer}}$ processes the flattened CNN features of all views together to output the scene representation:

$$S := \{s_i\}_{i=1}^{N_{\text{context}}{}^*(H/D)^*(W/D)} = E_{\text{transfomer}}\left(\{f_i\}_{i=1}^{N_{\text{context}}{}^*(H/D)^*(W/D)}\right) \tag{19}$$

where $\{f_i\}$ is the set of flattened CNN features.

**Rendering a view:** Given the scene representation $S$ and a target ray $r^*$, the decoder $D$ outputs an RGB pixel:

$$\hat{a}_{r^*} = D(\gamma(r^*), S) \in \mathbb{R}^3. \tag{20}$$

where $\gamma$ is the same function used in Eq. (18). The architecture of $D$ comprises two stacks of a cross-attention block followed by a feedforward MLP. The cross-attention layers determine which token in the set $S$ to attend to, to render a pixel corresponding to the given target ray. The output of the cross-attention layers is then processed by a 4-layer MLP, to get the final RGB prediction. The number of hidden dimensions of this MLP is set to $1536$.

**Optimization:** The encoder and the decoder are optimized by minimizing the mean squared error between given target pixels $a_r$ and the predictions:

$$\mathcal{L}(E, D) = \sum_{r^*} \|a_{r^*} - \hat{a}_{r^*}\|_2^2. \tag{21}$$

### C.2.2 Details of the architecture and loss of Du et al. (2023)

Du et al. (2023) proposes an SRT-based transformer NVS model with a sophisticated architecture. The major differences between their model and SRT are that they use a dense vision transformer (Ranftl et al., 2021) for their encoder. They also use an epipolar-based sampling technique to select context view tokens, a process that helps render pixels efficiently in the decoding process.

We use the same optimization losses for training models based on this architecture as Du et al. (2023). Specifically, we use the $L_1$ loss between target and predicted pixels on RealEstate10k and ACID. We also use the following combined loss after the 30K-th iterations on ACID.

$$L_1(P, \hat{P}) + \lambda_{\text{LPIPS}} L_{\text{LPIPS}}(P, \hat{P}) + \lambda_{\text{depth}} L_{\text{depth}}(P, \hat{P}) \tag{22}$$

where $P, P' \in \mathbb{R}^{32 \times 32 \times 3}$ are target and predicted patches. $L_{\text{LPIPS}}$ is the perceptual similarity metric proposed by Zhang et al. (2018). $L_{\text{depth}}$ is a regularization loss that promotes the smoothness of estimated depths in the model. Please refer to Du et al. (2023) for more details. On RealEstate10k, we found that using the combined loss above deteriorates reconstruction metrics. Therefore, we train models on RealEsatate10k solely with the $L_1$ loss for 300K iterations.

### C.2.3 APE- and RPE-based transformers on CLEVR-TR

For the APE-based model, we replace the ray embeddings in SRT with a linear projection of the combined 2D positional embedding and flattened $SE(3)$ matrix. To build an RPE-based model, we follow the same procedure as in Section 3.1 and apply the representations to the bias vectors appended to the QKV vectors. Each bias dimension is set to 16 for the $\sigma_{\text{cam}}$ and 16 for $\sigma_h$ and $\sigma_w$. The multiplicities and frequency parameters are determined as described in Section 3.2. $\{s, u, v\}$ is set to $\{4, 1, 1\}$ and $\{f\}$ is set to $\{1, ..., 1/2^3\}$ for both $\sigma_h$ and $\sigma_w$. Table 12 shows an extended version of Table 2, which includes LPIPS (Zhang et al., 2018) and SSIM performance.

### C.2.4 Implementation of other PE methods

**Frustum positional embeddings (Liu et al., 2022):** Given an intrinsic $K \in \mathbb{R}^{3 \times 3}$, we transform the 2D image position of each token by $K^{-1}(x, y, 1)^{\text{T}}$. We follow Liu et al. (2022) and generate points at multiple depths with the linear-increasing discretization (Reading et al., 2021), where each depth value at index $i = 1, ..., D$ is computed by

$$d_{min} + \frac{d_{max} - d_{min}}{D(D+1)} i(i+1) \tag{23}$$

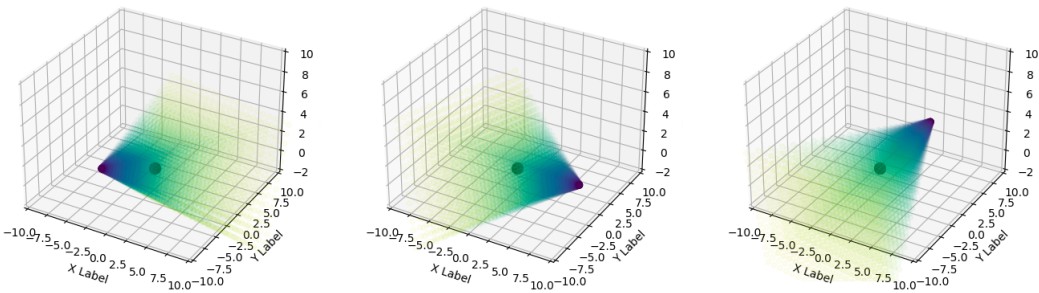

Fig. 16: **Frustum points on CLEVR-TR.** The black point indicates the origin $(0, 0, 0)$. Each object is sampled with its center in the range of $\in [-4, 4] \times [-4, 4] \times \{t/2\}$ where $t$ is the height of the object.

where $[d_{min}, d_{max}]$ is the full depth range and $D$ is the number of depth bins. Examples of the generated 3D points are visualized in Fig. 16. The concatenation of 3D points of multiple depth at each pixel is further processed by a learned 1-layer MLP, and added to input.

**Modulated layer normalization** (Hong et al., 2023; Liu et al., 2023a): Modulated layer normalization (MLN) modulates and biases each token feature $x$ by using vector features $\gamma, \beta$ each of which encodes geometric information. In Liu et al. (2023a), each token's geometric information is a triplet of a camera transformation, velocity, and time difference of consecutive frames. However, in our NVS tasks, the last two information does not exist. Thus, the vectors simply encode the camera transformation. Each $\gamma, \beta$ is computed by: $\gamma = \xi_\gamma(vec(E^{-1})), \beta = \xi_\beta(vec(E^{-1}))$ where $vec$ flattens the input matrix and $\xi_{\gamma, \beta}$ are learned linear transformations. Each token $x$ is transformed with $\gamma$ and $\beta$ as follows:

$$x' = \gamma \odot LN(x) + \beta \tag{24}$$

where $\odot$ denotes element-wise multiplication.

**Geometry-biased transformers (GBT)** (Venkat et al., 2023): GBT biases the attention matrix of each layer by using the ray distance. Specifically, suppose each token associates with a ray $r = (o, d) \in \mathbb{R}^3 \times \mathcal{S}^2$. GBT first converts $r$ into plücker coordinate $r' = (d, m)$ where $m = o \times d$. Then the ray distance between two rays $r'^q = (d^q, m^q)$ and $r'^k = (d^k, m^k)$ linked to each query vector $q$ and key vector $k$ is computed by:

$$dist(r'^q, r'^k) = \begin{cases} \frac{|d^q \cdot m^k + d^k \cdot m^q|}{||d_q \times d_k||_2} & \text{if } d^q \times d^k \neq 0 \\ \frac{||d^q(m^q - m^k/s) + d^k||_2}{||d^q||_2^2} & \text{if } d^q = sd^k, s \neq 0. \end{cases} \tag{25}$$

The GBT's attention matrix is computed by:

$$\mathrm{softmax}(QK^{\mathrm{T}} - \gamma^2 D(Q, K)), \tag{26}$$

where $D(Q, K) \in \mathbb{R}^{N \times N}, D_{ij}(Q, K) = dist(r'^{Q_i}, r'^{K_j})$. $\gamma \in \mathbb{R}$ is a learned scaler parameter that controls the magnitude of the distance bias. Following Venkat et al. (2023), in addition to this bias term, we also add a Fourier positional embedding computed with the plücker coordinate representation of the ray at each patch in the encoder and at each pixel in the decoder.

**Element-wise multiplication:** In this approach, for each token with a geometric attribute $g$, we first concatenate the flattened $SE(3)$ homogeneous matrix and flattened $SO(2)$ image positional representations with multiple frequencies. The number of frequencies is set to the same number as in GTA on CLEVR-TR. The concatenated flattened matrices are then linearly transformed to the same dimensional vectors as each $Q, K, V$. Then these vectors are element-wise multiplied to $Q, K, V$ and the output of $Attn$ in Eq. (6) in a similar way to GTA.

**RoPE+FTL** (Su et al., 2021; Worrall et al., 2017): RoPE (Su et al., 2021) is similar to GTA but does not use the SE(3) part (extrinsic matrices) as well as transformations on value vectors. In this approach, we remove SE(3) component from the representations. Also, we remove the transformations on the value vectors from each attention layer. As an implementation of FTL (Worrall et al.,

2017), we apply SE(3) matrices to the encoder output to get transformed features to render novel views with the decoder.

### C.3 2D IMAGE GENERATION WITH DiT (PEEBLES & XIE, 2023)

RoPE (Su et al., 2021) is a method commonly used to encode positional information in transformer models. GTA and RoPE are similar but differ in that, in GTA, group transformations are applied to the value vectors in addition to the query and key vectors, leading to improvements in our NVS tasks compared to models without this transformation. To further investigate the effectiveness of the value transformation, we conduct a 2D image generation experiment. We will describe the experimental setting in the following. We also opensource the code for this experiments in the same repository as our NVS experiments, and please refer to it for further details.

Following the experimental setup of DiT (Peebles & Xie, 2023), we use a transformer-based denoising network for image generation on ImageNet (Russakovsky et al., 2015). The image resolution is set to 256x256, and we choose the DiT-B/2 model as our baseline. Since the original DiT model does not adopt RoPE encoding, we trained models with both RoPE and GTA positional encodings. We use the same representation matrix $\rho_g$ for both RoPE and GTA, which is written as follows:

$$\rho_g := \sigma_h(\theta_h) \oplus \sigma_w(\theta_w). \tag{27}$$

Here, the notation of each symbol is the same as in the main section. The representation design of each $\sigma_h$ and $\sigma_w$ follows the original work of RoPE (Su et al., 2021). Training of each model is conducted for 2.5M iterations (approximately 500 epochs) with batch size of 256. We experiment with mixed-precision training (BFloat16), but observed instability when using RoPE and GTA. To address this, we adopt RMSNorm (Zhang & Sennrich, 2019) applied to each $Q$ and $K$ vector, with which we find that no instability is made throughout the training. We report in Table 5 (Right) inception scores and FIDs with classifier-free guidance and its scale set to 1.5. We show comparisons of generated images in Section E.

# D  RENDERED IMAGES

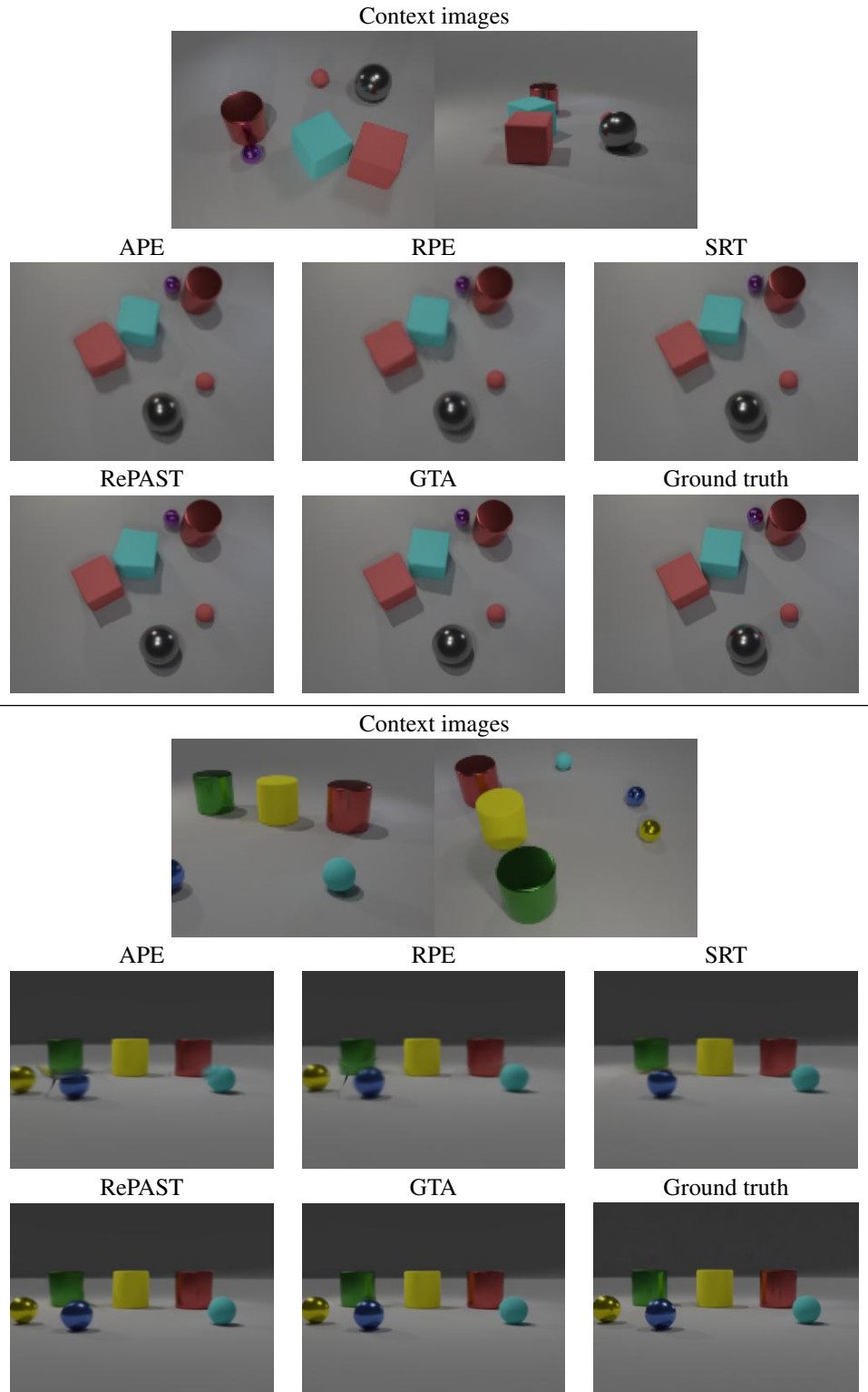

Fig. 17: **Qualitative results on CLEVR-TR.**

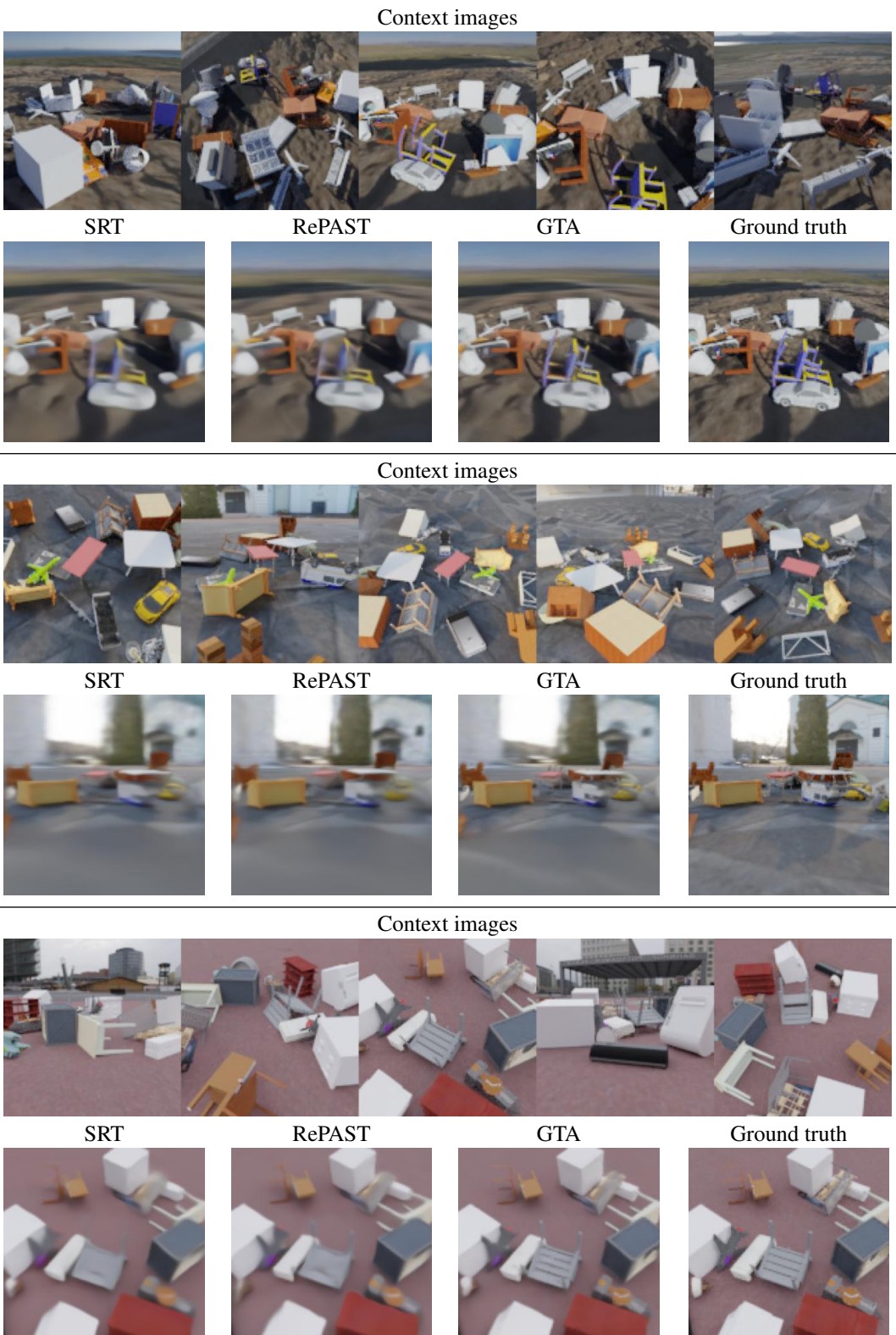

Fig. 18: **Qualitative results on MSN-Hard.**

| Context images | Du et al. (2023) | GTA | Ground truth |
|---|---|---|---|

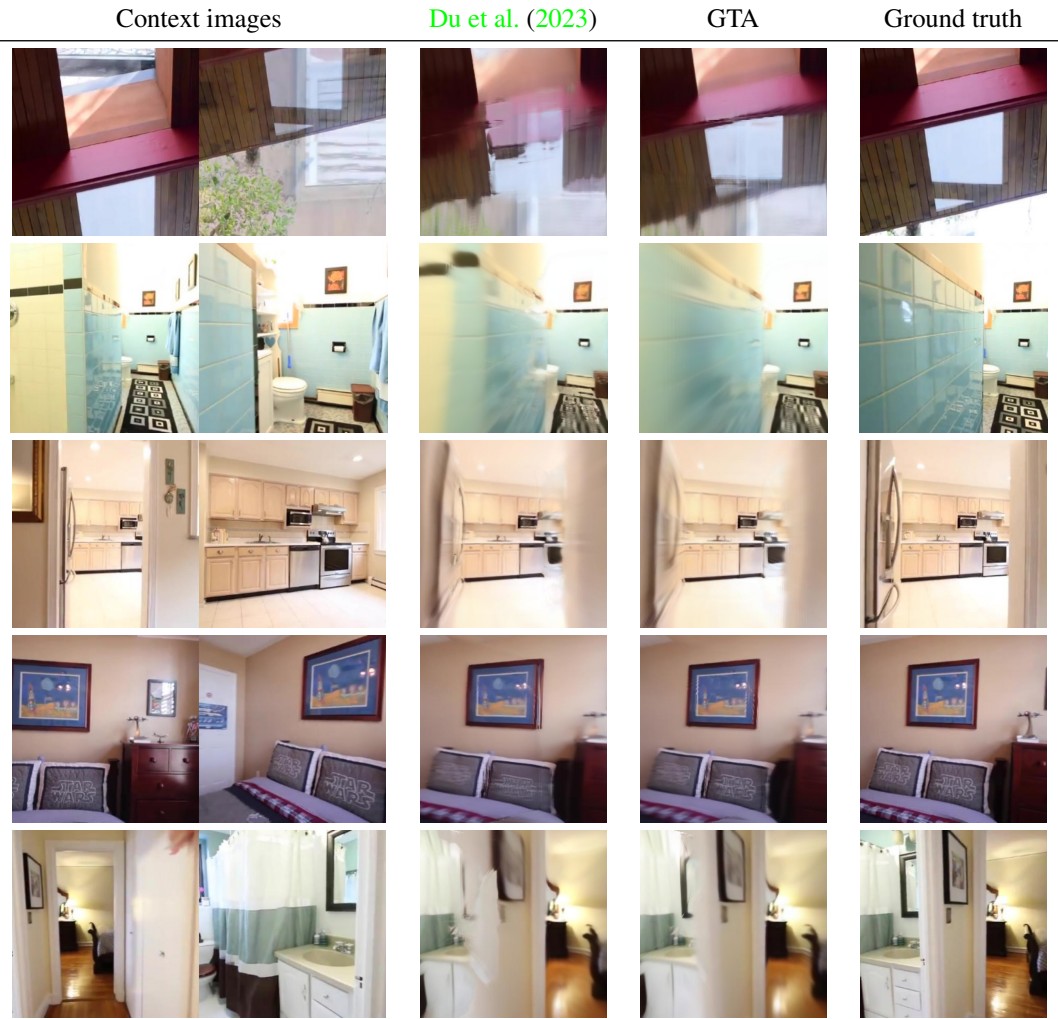

Fig. 19: **Qualitative results on RealEstate10k.**

| Context images | Du et al. (2023) | GTA | Ground truth |
| --- | --- | --- | --- |

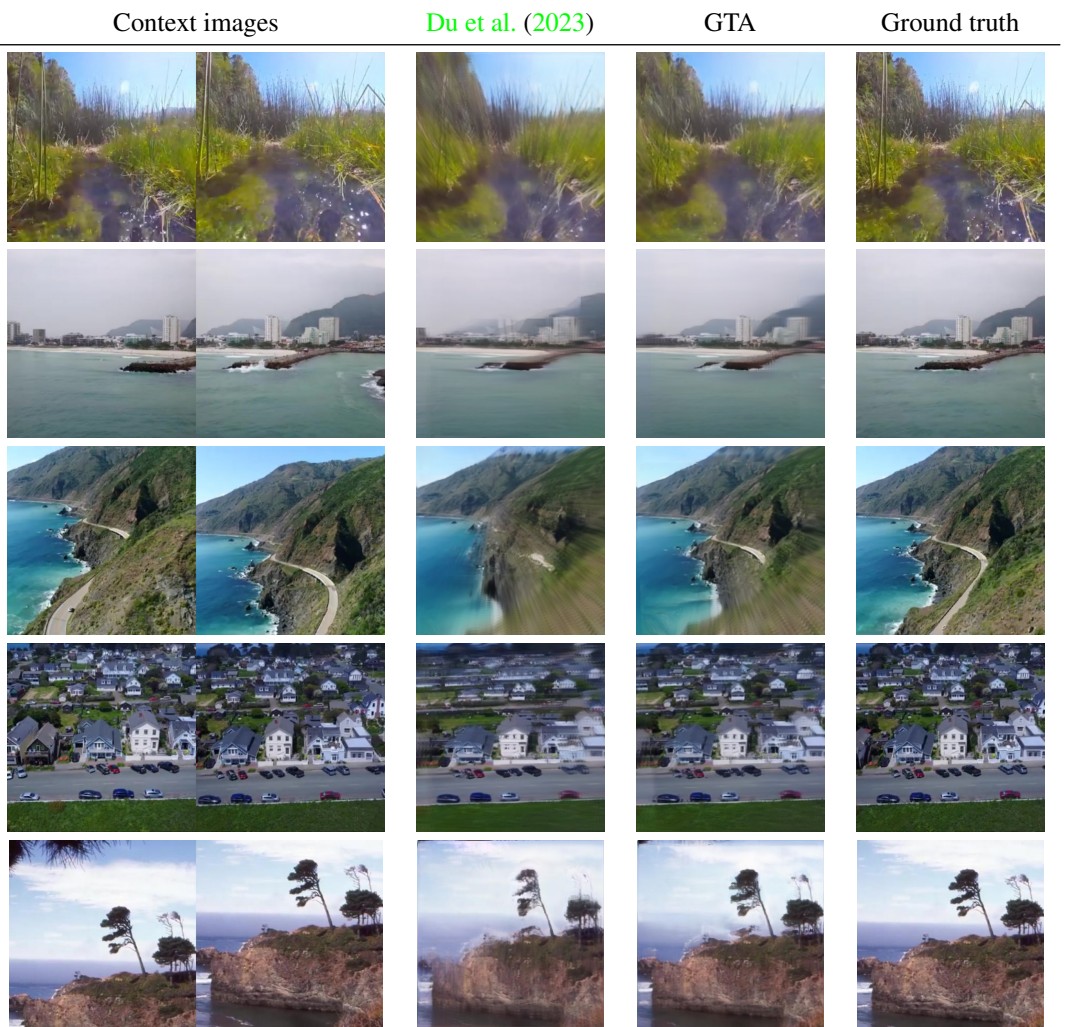

Fig. 20: **Qualitative results on ACID.**

# E    GENERATED IMAGES OF DITS

DiT + GTA          DiT + RoPE          DiT (Peebles & Xie, 2023)

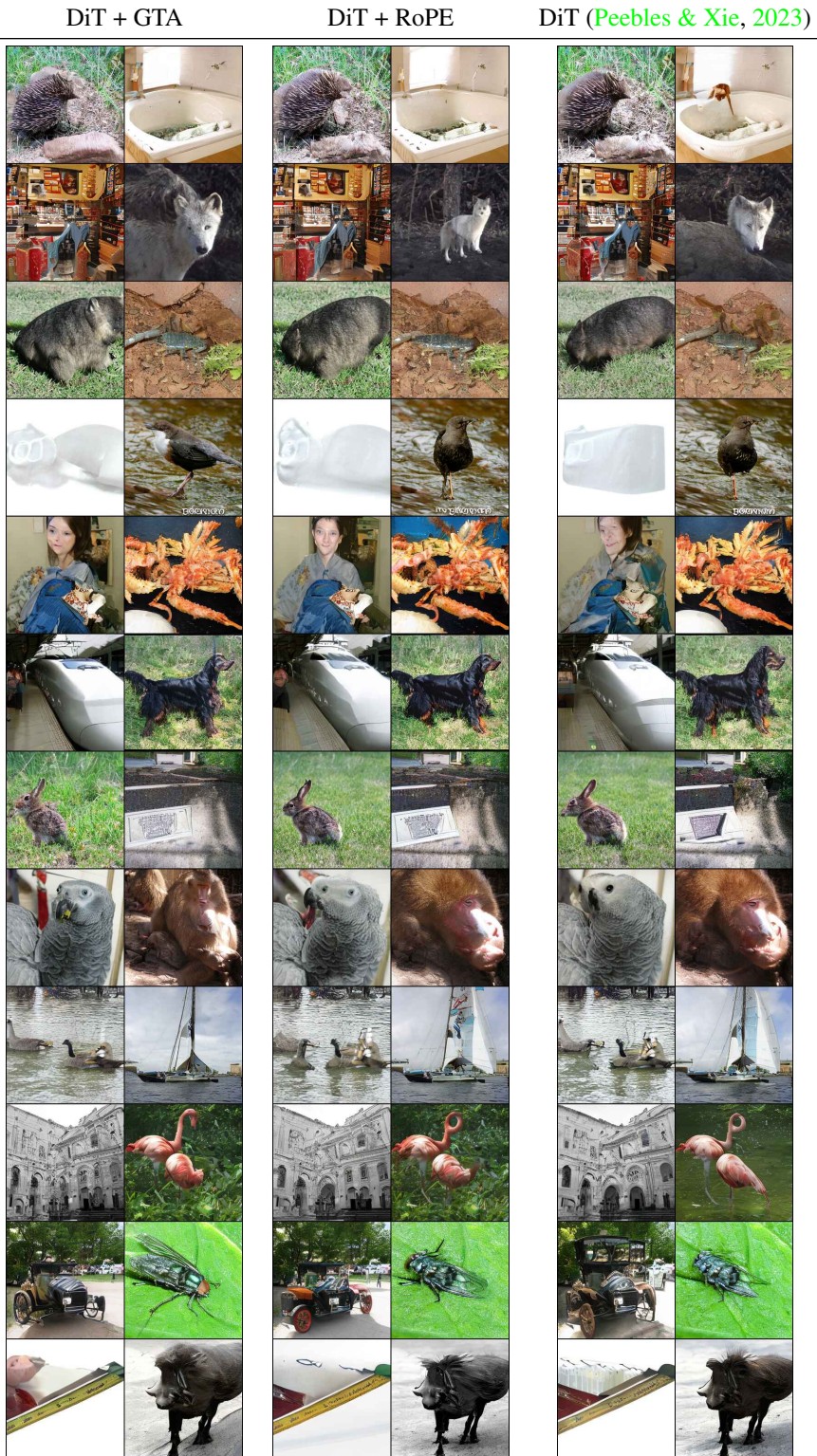

Fig. 21: **Class-conditional generation on ImageNet**. Labels and noises are randomly sampled.

DiT + GTA  DiT + RoPE  DiT (Peebles & Xie, 2023)

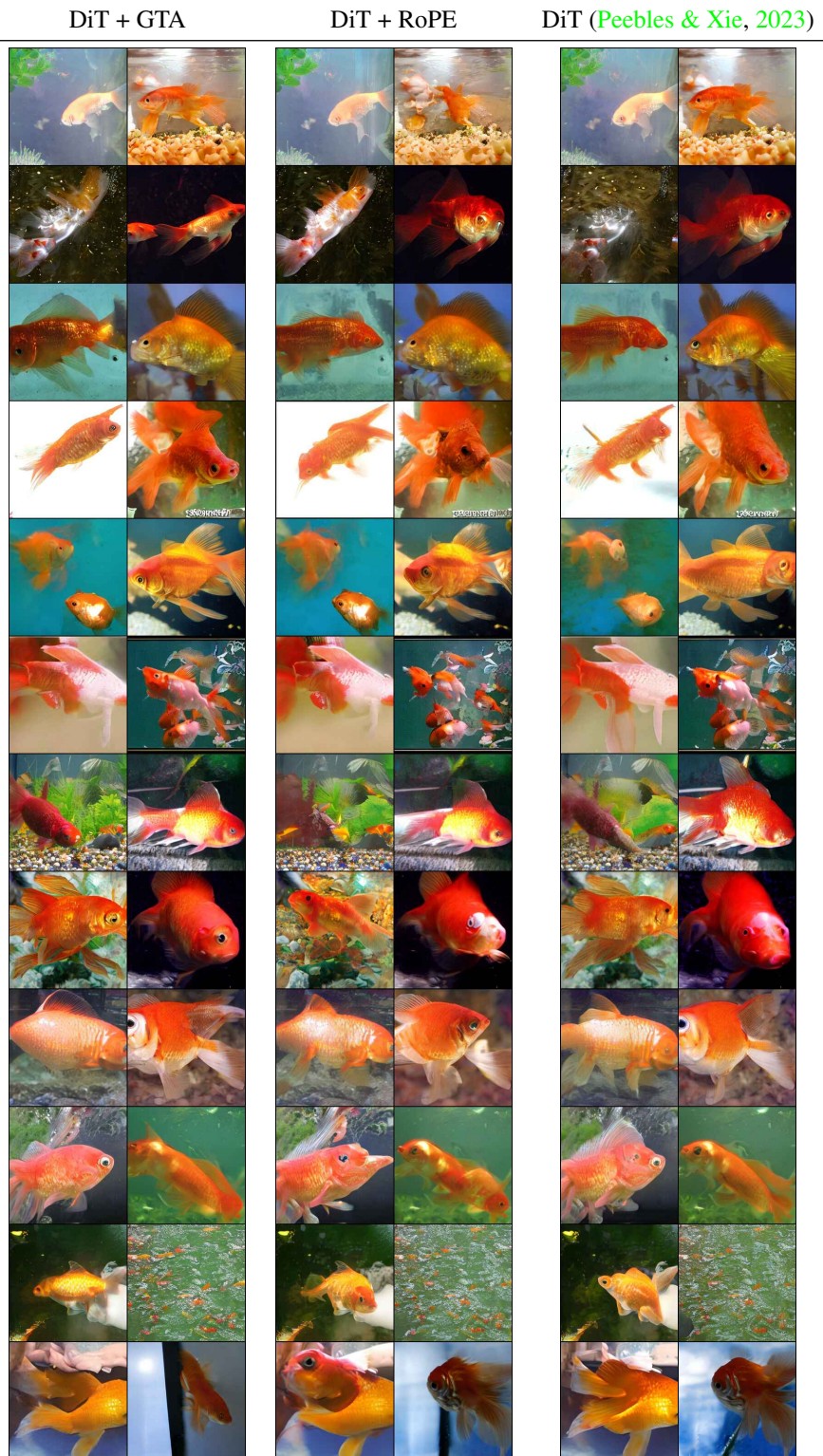

Fig. 22: **Generated images with class label 'Goldfish'**

DiT + GTA    DiT + RoPE    DiT (Peebles & Xie, 2023)

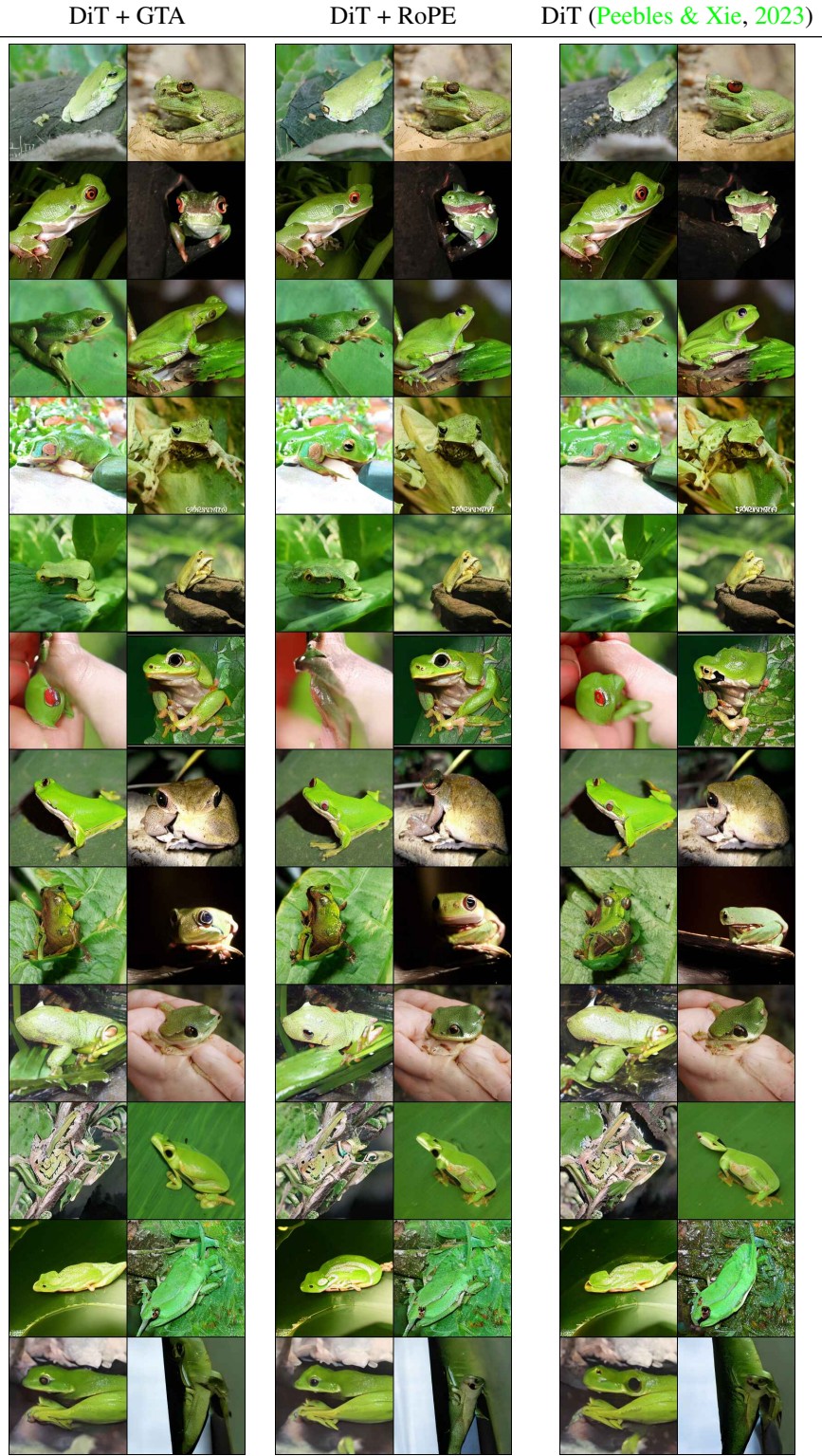

Fig. 23: **Generated images with class label 'Tree frog'**

DiT + GTA          DiT + RoPE          DiT (Peebles & Xie, 2023)

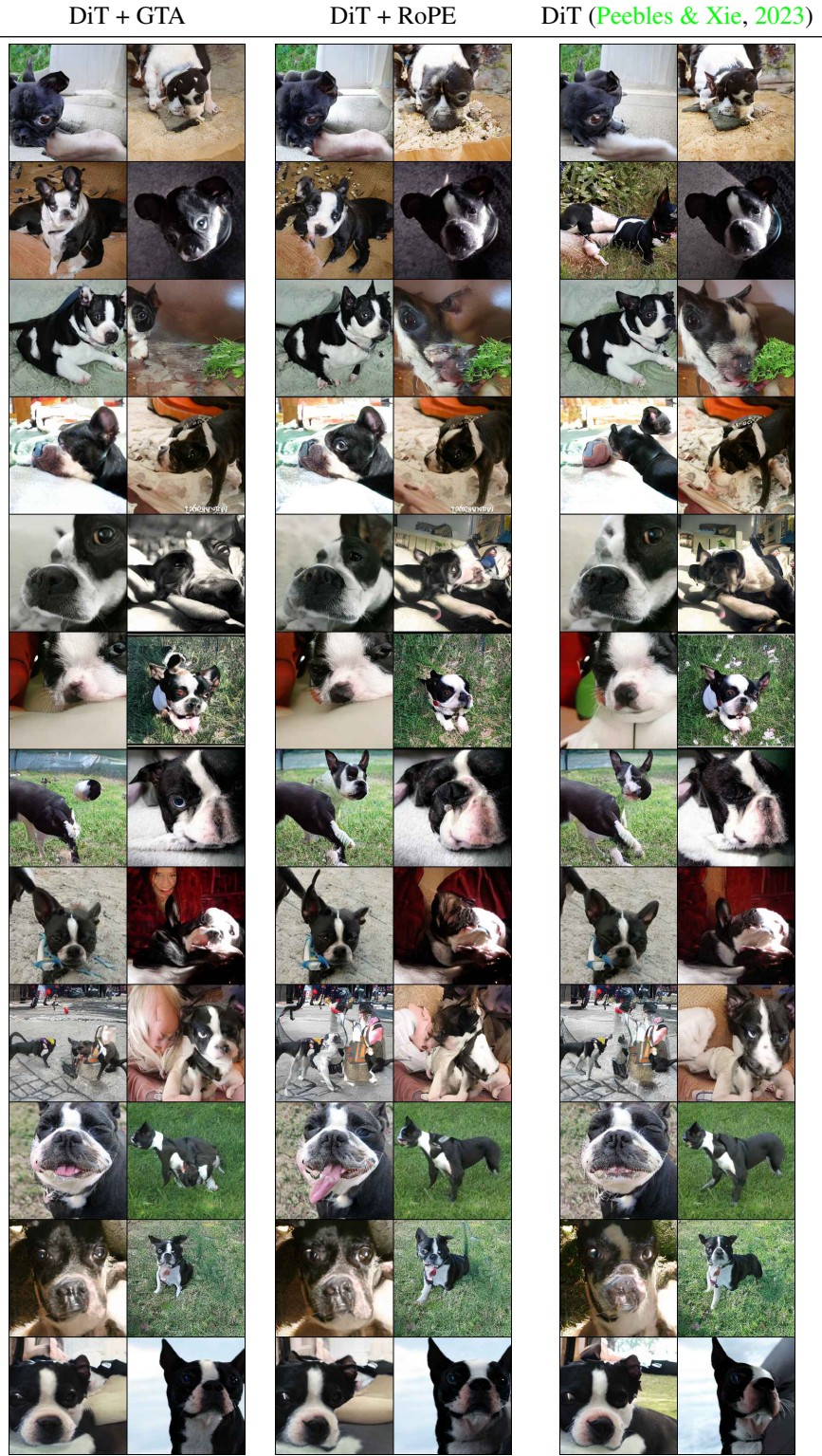

Fig. 24: **Generated images with label 'Boston bull'**

DiT + GTA          DiT + RoPE          DiT (Peebles & Xie, 2023)

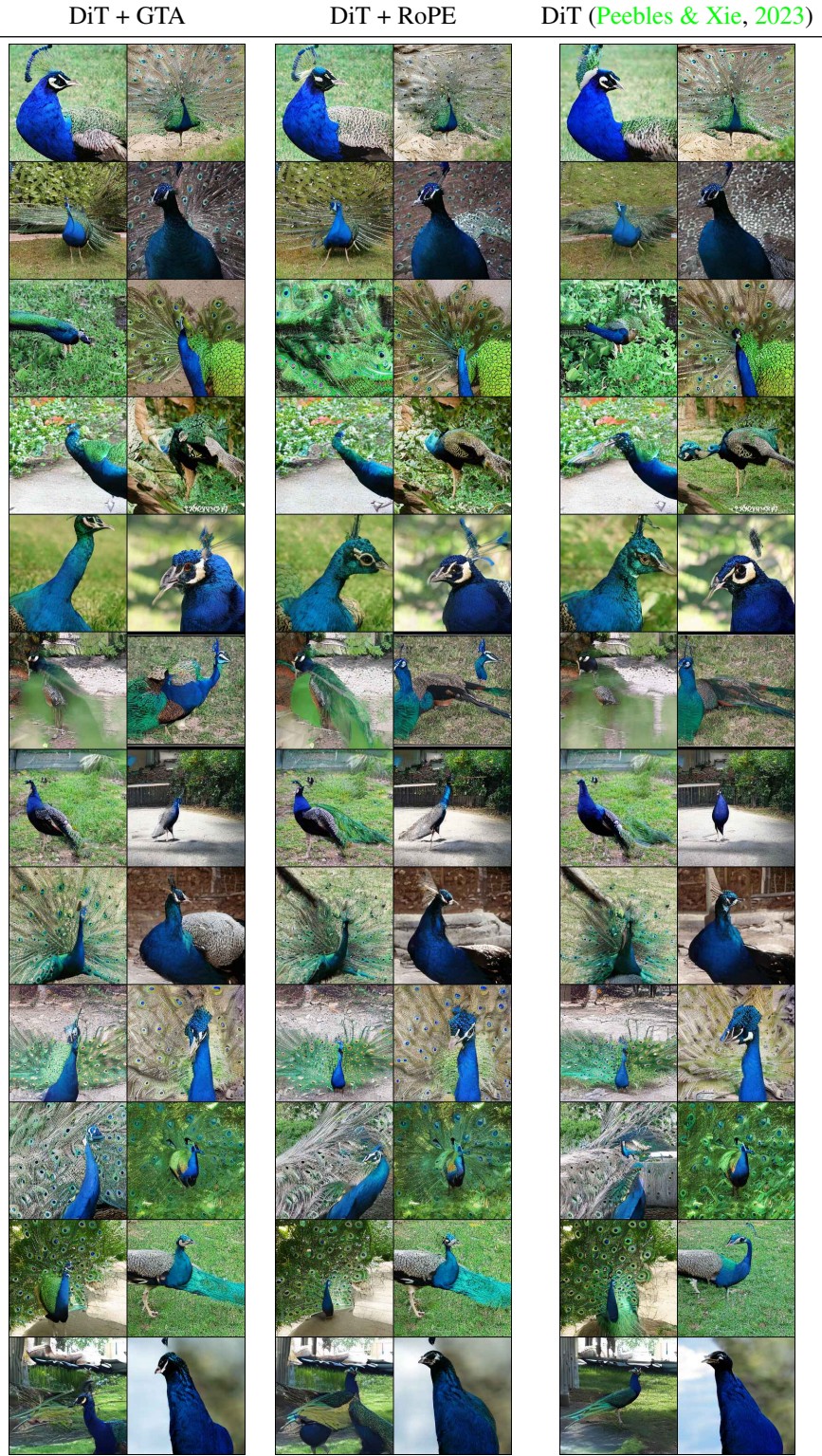

Fig. 25: **Generated images with label 'Peacock'**

