# OpenReview forum: "GTA: A Geometry-Aware Attention Mechanism for Multi-View Transformers"
_ICLR.cc/2024/Conference — ICLR 2024 poster_

### Official Review · Reviewer_nfN9 · 2023-10-17

**Soundness:** 3 good
**Presentation:** 3 good
**Contribution:** 3 good
**Rating:** 6
**Confidence:** 4

**Summary:**

This paper focuses on enhancing the attention mechanism by incorporating geometric information. The motivation behind this is the observation that existing positional encoding techniques are not optimally suited for vision tasks. In particular, the authors encode the geometric structure of tokens by representing them as relative transformations and introduce a novel geometric-aware attention mechanism. Experiments are conducted on novel view synthesis tasks, where the proposed method achieves state-of-the-art performance among transformer-based neural video synthesis (NVS) methods.

**Strengths:**

1. This paper introduces a novel attention mechanism grounded in geometric information, and the motivation is reasonable.
2. The paper is well-written and the experiments yield compelling qualitative results compared to baseline models.

**Weaknesses:**

- Lack of comparisons with the latest NeRF-based models, except Pixel-NeRF which is proposed in 2021.

**Questions:**

The resolution of training images used in the experiment seems to be not high. How does this method perform on high-resolution data, and what is the impact of different numbers of training views?

---

> ### Author Response · Authors · 2023-11-12
> **Clarification on 'different numbers of training views' in your question**
>
> Thank you very much for your feedback.  In response to your concerns, we are planning to conduct additional experiments and would like to clarify one thing in your question: Regarding 'different numbers of training views', does this mean different numbers of training *scenes*, or does it refer to different numbers of *input context views* for each individual scene?

---

> > ### Comment · Reviewer_nfN9 · 2023-11-13
> >
> > I mean the different numbers of input views for each scene.

---

> ### Author Response · Authors · 2023-11-22
> **Response to reviewer nfN9**
>
> Thank you very much for your feedback. To address your concerns, we conducted additional experiments on RealEstate10K.
>
> >Lack of comparisons with the latest NeRF-based models, except Pixel-NeRF which is proposed in 2021.
>
> We added additional NeRF-based models from the Appendix of Du et al. 2023 into Table 3.
> Additionally, we trained and evaluated the latest concurrent NeRF-based approach, MatchNeRF, from arXiv on RealEstate10k. We included these numbers in Table 3 in the revised manuscript.
>
> | Model                              	| PSNR  | LPIPS | SSIM   |
> |----------------------------------------|:-------:|:-------:|:-------:|
> | PixelNeRF (Yu et al. 2021)         	| 13.91 | 0.591 | 0.460 |
> | StereoNeRF (Chibane et al. 2021)   	| 15.40 | 0.604 | 0.486 |
> | IBRNet (Wang et al. 2021)          	| 15.99 | 0.532 | 0.484 |
> | GeoNeRF (Johari et al. 2022)       	| 16.65 | 0.541 | 0.511 |
> | MatchNeRF (Chen et al. 2023)       	| **23.06** | 0.258 | 0.830 |
> | -------------------------------------- |-------|-------|-------|
> | GPNR (Suhail et al. 2022)          	| 18.55 | 0.459 | 0.748 |
> | (Du et al. 2023)                   	| 21.65 | 0.285 | 0.822 |
> | (Du et al. 2023) + GTA (Ours)     | 22.85 | **0.255** | **0.850** |
>
> We grouped the works into methods using NeRF-based rendering and methods using Transformer-based rendering. We find that MatchNeRF outperforms both NeRF-based and Transformer-based prior work by large margins. By combining our GTA mechanism with Du et al. 2023, the transformer-based method also achieves state-of-the-art results.
> We would like to mention that the transformer-based NVS with GTA, which introduces a relatively weaker prior into the model, is competitive with the model equipped with a strong physical prior (volumetric rendering).  Our attention mechanism effectively steers the transformer to learn the underlying 3D information behind images.
>
>
>
> >How does this method perform on high-resolution data, …
>
> We conducted experiments on RealEstate10k with 384x384 resolutions (1.5 times larger in height and width than in Table 3). We see that GTA also improves over the baseline model at higher resolution. We added this table as Table 7.
>
> |Method| PSNR  | LPIPS 	| SSIM |
> |----|:-----------:|:-----------:|:-----------:|
> | Du et al. 2023 | 21.29 | 0.341| 0.838 |
> | Du et al. 2023 + GTA (ours)| **21.95** | **0.322** | **0.849** |
>
> Due to limited computational resources and discussion period, we only trained for 100K iterations, which is one-third of the iterations in Table 3. We will include results with fully trained models in the final version.
>
> >...what is the impact of different numbers of training views?
>
> We trained models with 3-context views and show the results below: We see GTA improves reconstruction quality over Du et al. 2023 with 3 views setting. We added this table as Table 8.
>
> | Method|  2-views PSNR | 3-views PSNR      |
> |----------------|:-----------------:|:--------------:|
> | (Du et al. 2023) |  	21.43  |            21.84      |
> | (Du et al. 2023) + GTA (ours)     | 	**22.17** 	|         **22.50**    |
>
> Also in this table, numbers are computed with models trained for 100K iterations, and we will include results with fully trained models in the final version.
>
>
> References
> * Alex Yu, Vickie Ye, Matthew Tancik, and Angjoo Kanazawa. PixelNeRF: Neural radiance fields from one or few images. In Proc. IEEE Conf. on Computer Vision and Pattern Recognition (CVPR), 2021
> * Julian Chibane, Aayush Bansal, Verica Lazova, and Gerard Pons-Moll. Stereo radiance fields (SRF): learning view synthesis for sparse views of novel scenes. In Proc. IEEE Conf. on Computer Vision and Pattern Recognition (CVPR), 2021.
> * Qianqian Wang, Zhicheng Wang, Kyle Genova, Pratul P Srinivasan, Howard Zhou, Jonathan T Barron, Ricardo Martin-Brualla, Noah Snavely, and Thomas Funkhouser. IBRNet: Learning multi-view image-based rendering. In Proc. IEEE Conf. on Computer Vision and Pattern Recognition (CVPR), 2021
> * Mohammad Mahdi Johari, Yann Lepoittevin, and Franc ̧ois Fleuret. Geonerf: Generalizing nerf with geometry priors. In Proc. IEEE Conf. on Computer Vision and Pattern Recognition (CVPR), 2022.
> * Yuedong Chen, Haofei Xu, Qianyi Wu, Chuanxia Zheng, Tat-Jen Cham, and Jianfei Cai. Explicit correspondence matching for generalizable neural radiance fields. arXiv.org, 2023.
> * Yilun Du, Cameron Smith, Ayush Tewari, and Vincent Sitzmann. Learning to render novel views from wide-baseline stereo pairs. In Proc. IEEE Conf. on Computer Vision and Pattern Recognition (CVPR), 2023

---

### Official Review · Reviewer_LyNT · 2023-10-29

**Soundness:** 3 good
**Presentation:** 2 fair
**Contribution:** 2 fair
**Rating:** 5
**Confidence:** 3

**Summary:**

This paper addresses the limitation of existing positional encoding designs in 3D vision tasks, which do not consider task-relevant 3D geometric structures. The authors propose a geometry-aware attention mechanism that encodes the geometric structure of tokens for the novel view synthesis task. Extensive experiments demonstrate the effectiveness of the proposed method, and qualitative comparisons with previous state-of-the-art techniques highlight significant improvements.

**Strengths:**

1. The proposed GTA consistently outperforms existing state-of-the-art methods on multiple datasets, including RealEstate10k, ACID, CLEVR-TR, and MSN-Hard.

2. The authors provide extensive qualitative comparisons and analyses, effectively supporting the superiority of the proposed method over previous state-of-the-art techniques.

**Weaknesses:**

1. This paper's organization could be improved. For instance, placing the related work (Section 5) after the introduction would help readers understand the difference between conventional positional encoding schemes and the proposed relative transformation encoding scheme. The authors should summarize existing geometric-related positional encoding techniques (not only in novel view synthesis, but also in many 3D vision tasks), and conduct a comprehensive comparison. Since the background information of transformer (the first half of section 2 Background) is already well-known, it can be moved to the appendix.

2. The paper's contribution appears to be incremental, offering limited insights from the proposed relative transformation positional encoding alone.

3. While the paper introduces GTA as a novel positional encoding scheme, it seems to function more like a transformer block (consisting of positional encoding and the attention block) than just a positional encoding scheme. A more detailed ablation study of each component within GTA is needed.

In general, improving the paper's organization and conducting a more comprehensive comparison with existing techniques would strengthen its contributions. Additionally, more detailed analysis of the components within GTA would enhance the paper's overall quality.

**Questions:**

see weakness

---

> ### Author Response · Authors · 2023-11-12
> **Clarification on ablation studies**
>
> Thank you very much for your feedback. We plan to conduct additional experiments in response to your requests.
>
> Regarding Weakness 3, where it is mentioned:
> >A more detailed ablation study of each component within GTA is needed.
>
> Could you please clarify what specific ablation studies you are envisioning? In Table 5, we have already demonstrated the results when removing the representation matrix applied to the value vector in Equation (4). Additionally, Table 4 shows the results using different representation matrices. Could you suggest what other types of ablation experiments you think would be beneficial?

---

> > ### Comment · Reviewer_LyNT · 2023-11-16
> >
> > I mean with or without relative transformation positional encoding. While Table 2 presents the performance comparison of your approach with PRE/APE, these two are the most basic encoding approaches. As I mentioned, it would be better to summarize existing geometric-related positional encoding techniques (not only in novel view synthesis, but also in many 3D vision tasks), and conduct a comprehensive comparison with these approaches.

---

> ### Author Response · Authors · 2023-11-22
> **Response to reviewer LyNT**
>
> Thank you very much for your feedback. In the revised version, we add a number of ablation studies and numerical comparisons to existing positional encoding schemes in 3D visions. We hope our new version of the manuscript and the response below will address your concerns.
>
> >This paper's organization could be improved. For instance, placing the related work (Section 5) after the introduction would help readers understand the difference between conventional positional encoding schemes and the proposed relative transformation encoding scheme.
>
> Thank you for your feedback and suggestions. As suggested, in the revised manuscript we put discussions of 3D position encodings in Section 2 and renamed it to Related work.
>
> > The authors should summarize existing geometric-related positional encoding techniques (not only in novel view synthesis, but also in many 3D vision tasks), and conduct a comprehensive comparison
>
> In Related work, we added discussions of existing positional encoding (PE) schemes from other 3D vision communities besides NVS. Specifically, we include PEs used in state-of-the-art 3D object detection and segmentation tasks (Frustum embedding, Motion Layer normalization), PEs when depth supervision is available (3DPPE), and also we also include a recent PE method used in geometry-biased transformer (GBT), which biases the attention matrix of each layer by ray distance between each query and key pair. Also we added 3D positional embedding methods for point clouds.
>
> >Background info on TransFormer can be moved to the appendix.
>
> Thank you for the suggestion. We make the transformer’s background more concise. We keep the introductions of APE and RPE in Section 2, since it should be helpful for readers to easily understand the difference to our proposed method.
>
> > A more detailed ablation study of each component within GTA is needed.
>
> (Response to our question)
> > I mean with or without relative transformation positional encoding. While Table 2 presents the performance comparison of your approach with PRE/APE, these two are the most basic encoding approaches. As I mentioned, it would be better to summarize existing geometric-related positional encoding techniques (not only in novel view synthesis, but also in many 3D vision tasks), and conduct a comprehensive comparison with these approaches
>
> Thank you for your suggestions. We conducted additional experiments with various PEs. The below table shows test PSNRs on CLEVR-TR with Frustum Embedding (Liu et al. 2022), Motion layer norm (MLN) (Liu et al. 2023), Elementwise Mul., RoPE (Su et al. 2021) + FTL (Worrall et al. 2017) and GBT (Venkat et al. 2023). We observe that GTA achieves the best PSNR. This table was added to the revised manuscript as Table 5.
>
> | Method                                    	| PSNR  	|
> |-----------------------------------------------|:-----------:|
> | MLN (Liu et al. 2023)                     	| 31.97    	|
> | Element-wise Mul.                         	| 34.33   	|
> | GBT (Venkat et al. 2023)                  	| 35.63  	|
> | Frustum Embedding (Liu et al. 2022)   | 37.23 |
> | RoPE+FTL (Su et al. 2021, Worrall et al. 2017) | 38.18   	|
> | GTA                                       	| **38.99** |
>
> Experimental settings for this comparison are described in the Appendix E.4.

---

> > ### Author Response · Authors · 2023-11-22
> >
> > > The paper's contribution appears to be incremental, offering limited insights from the proposed relative transformation positional encoding alone.
> >
> > As we describe in the related work section, most of the existing works of transformers use positional encoding methods that do not faithfully keep the geometric structure associated with tokens. We posit that the structure-preserving way to encode geometric information of tokens; i.e.) the homomorphism property of the transformations applied to QKV vectors is important. The homomorphic transformation makes features more structurally position-aware, and therefore, subsequent layers easily learn the effect of the positional encoding, which is supported by Figure 6, showing GTA significantly improves learning efficiency over SRT and RePAST.
> > The element-wise multiplication experiment requested by reviewer ZuJ5 additionally supports this. In that experiment, we compare GTA against element-wise multiplication of features that encode geometric information. When the representations are transformed into vector form multiplied to QKV vectors, the homomorphism property is lost. Suppose $f(g) :=W vec(\rho(g))$ is the vector embedding of the representations $\rho(g)$, then $f(gg’) \neq f(g)f(g’)$ in general. Thus, f is not faithful in terms of structure preservation of $g$. We also see in the table below that the performance significantly degrades when using element-wise multiplication to the QKV features.
> > | Method                  	|	PSNR 	|
> > |-----------------------------|:-----------:|
> > | Element-wise multiplication |  34.33  |
> > | GTA (ours)              	|  **38.99**  |
> >
> > See also the response to reviewer myD9 regarding the importance of geometry-preserving positional encoding of GTA.
> >
> > RoPE (Su et al. 2021) is similar to GTA. In our manuscript, we broaden the scope of the transformations from just 1D sequence in RoPE to geometric structures such as SE(3) (camera transformations in our experiments) and can cover even more general geometric structures as long as they can be defined in terms of a group. We also form an intuitive explanation in Section 3. The transformation on V, which is missing in RoPE, is found to be important, and Table 4(c) shows removing transformations on V degrades performance significantly. This is expected from the intuition behind GTA. Value vectors, which will be added to other token features from different coordinate systems, should also be transformed with the relative transformation, so that the addition is performed in an aligned coordinate space.
> >
> > References
> > * Yingfei Liu, Tiancai Wang, Xiangyu Zhang, and Jian Sun. Petr: Position embedding transformation for multiview 3d object detection. In Proc. of the European Conf. on Computer Vision (ECCV), 2022.
> > * Ruoshi Liu, Rundi Wu, Basile Van Hoorick, Pavel Tokmakov, Sergey Zakharov, and Carl Vondrick. Zero-1-to-3: Zero-shot one image to 3d object. arXiv.org, 2023.
> > * Naveen Venkat, Mayank Agarwal, Maneesh Singh, and Shubham Tulsiani. Geometry-biased transformers for novel view synthesis. arXiv.org, 2023.
> > * Jianlin Su, Yu Lu, Shengfeng Pan, Ahmed Murtadha, Bo Wen, and Yunfeng Liu. Roformer: Enhanced transformer with rotary position embedding. arXiv.org, 2021.
> > * Daniel E Worrall, Stephan J Garbin, Daniyar Turmukhambetov, and Gabriel J Brostow. Interpretable transformations with encoder-decoder networks. In Proc. of the IEEE International Conf. on Computer Vision (ICCV), 2017.

---

### Official Review · Reviewer_ZuJ5 · 2023-10-31

**Soundness:** 3 good
**Presentation:** 2 fair
**Contribution:** 3 good
**Rating:** 6
**Confidence:** 3

**Summary:**

The paper proposes a new methodology for encoding positional information and camera transformations in 3D vision tasks. It employs a special Euclidean group to encode the geometric relationship between queries and key-value pairs within the attention mechanism. Experimental results demonstrate the effectiveness and efficiency of the proposed method, achieving top performance on the novel view synthesis dataset.

**Strengths:**

1. The geometric relationship is crucial for multi-view geometry tasks. The proposed method straightforwardly and reasonably encodes transformation and position information into the attention mechanism. The method has demonstrated superior results on several novel view synthesis datasets.
2. The involvement of the special Euclidean group in encoding transformations is interesting and useful for capturing geometric relationships.
3. The proposed method requires minimal computation, making it easy to implement and integrate.

**Weaknesses:**

1. Geometry-aware attention for transformers has been extensively explored, and the idea of encoding position and transformation information to improve 3D vision tasks has been discussed previously. While the detailed implementation of the encoding process (such as applying transformations to value vectors) differs from existing works, they share a similar underlying concept, thereby diminishing the novelty of this paper.
2. The evaluation and ablation study primarily focus on the novel view synthesis task. However, since the proposed method is easily integrable into existing attention-based 3D vision tasks, I highly recommend validating the approach on other 3D vision tasks, such as multi-view 3D detection.
3. Unlike the feature warping approach, the encoded transformation information relies on matrix multiplication. To be honest, I don't fully grasp the physical meaning behind this.

**Questions:**

1. Table 4 validates that the method proposed in the paper effectively improves accuracy.  I am curious about the performance of other similar methods when evaluated against the same baseline and the same settings. Are they still inferior to the approach presented in the paper?
2. Additionally, since geometry-aware attention is a general method, can it also enhance accuracy in other perceptual tasks apart from novel-view synthesis?

---

> ### Author Response · Authors · 2023-11-12
> **Regarding Question 1**
>
> Thank you very much for your feedback. There is one thing we would like to clarify regarding Question 1.
> > I am curious about the performance of other similar methods when evaluated against the same baseline and the same settings. Are they still inferior to the approach presented in the paper?
>
> Could you clarify what is meant by 'other similar methods' in the response? Table 4 focuses on comparisons *within* the GTA mechanism. Specifically, Table 4(a) investigates the effects of removing each component of the representations (SE(3), SO(2), and SO(3)) on performance, while Table 4(b) assesses the impact of different representation choices for image pixel positional encoding on performance. Also, as a comparison to other positional encoding schemes, Table 2 on the left presents performance with absolute positional encoding (APE) and relative positional encoding (RPE). In both encoding strategies, flattened SE(3) and SO(2) matrices are added to the input of each attention layer. Here, we demonstrate the superior performance of our GTA-based model over models based on APE and RPE. Thus, we presume that 'other similar methods' might refer to comparable architectural approaches such as the architecture employed by Du. et al. 2023 and used for our experiments on RealEstate10k and ACID, and that your suggestion is to conduct the same experiments in Table 4 but with different architectures. However, we are not entirely confident in this interpretation. Could you please confirm if this understanding matches what you intended in your response?

---

> > ### Comment · Reviewer_ZuJ5 · 2023-11-15
> >
> > I mean other similar methods in image position encoding and camera encoding. For e.g., the rotary PE for image and FTL encoding for camera. Hear FTL~(https://openaccess.thecvf.com/content_ICCV_2017/papers/Worrall_Interpretable_Transformations_With_ICCV_2017_paper.pdf) and their following methods  also aims at encoding camera Intrinsics and extrinsics in transformers.
> > If we combine other  image PE and camera encoding methods, how much worse will the performance be compared to the method proposed in the paper?

---

> ### Author Response · Authors · 2023-11-22
> **Response to reviewer ZuJ5**
>
> Thank you very much for your feedback. We respond to your concerns one by one below.
>
> >While the detailed implementation of the encoding process (such as applying transformations to value vectors) differs from existing works, they share a similar underlying concept
>
> In our manuscript, we define our GTA in the language of group theory, which enables us to broaden the scope of the transformations from just 1D sequence in existing work (RoPE) to geometric structures such as SE(3) (camera transformations in our experiments) and can cover even more general geometric structures as long as they can be defined in terms of a group.
> Also, the transformation on V, which is missing in RoPE, is not a heuristic in GTA, but naturally arises in our formulation with an intuitive explanation. Value vectors, which will be added to other token features from different coordinate systems, should also be transformed with the relative transformation so that the addition is performed in an aligned coordinate space. Table 4(c) shows removing transformations on V degrades performance significantly.
>
> >Table 4 validates that the method proposed in the paper effectively improves accuracy. I am curious about the performance of other similar methods when evaluated against the same baseline and the same settings. Are they still inferior to the approach presented in the paper?
>
> (From the response to our question)
> >  I mean other similar methods in image position encoding and camera encoding. For e.g., the rotary PE for image and FTL encoding for camera.
>
> Thank you for your questions.  To address your concern, we tested with a RoPE  (Su et al. 2021) + FTL (Worrall et al. 2017) model on CLEVR-TR. RoPE is similar to GTA but does not use the SE(3) part (extrinsic matrices) as well as transformations on value vectors. FTL is applying SE(3) matrices to the encoder output to get transformed features to render novel views with the decoder. Note that this is a similar method to the second method in Table 4(a) where we remove SE(3) representation matrices from the encoder. The results are below, and we see the performance of RoPE + FTL is inferior to GTA and also found to be similar to the performance of GTA without SE(3) in the encoder, as expected. We included the number of RoPE+FTL in Table 5.
> | Method                                        	|	PSNR 	|
> |---------------------------------------------------|:-----------:|
> | RoPE (Su et al. 2021) + FTL (Worrall et al. 2017) | 	38.18  	|
> | GTA (ours)                                    	|  **38.99**  |
>
> >Since the proposed method is easily integrated into existing attention-based 3D vision tasks, I highly recommend validating the approach on other 3D vision tasks, such as multi-view 3D detection.
> >Additionally, since geometry-aware attention is a general method, can it also enhance accuracy in other perceptual tasks apart from novel-view synthesis?
>
> Indeed, the general idea of GTA is applicable to other 3D vision tasks.
> In 3D detection and segmentation, the target coordinate system is typically an orthographic birds eye view (BEV) grid which requires a different $\rho$ for the decoder. Additionally, methods in these communities work with high-resolution feature grids, requiring sparse attention mechanisms like deformable attention to work efficiently. We think extending GTA to sparse attention and designing a novel representation $\rho$ for orthographic target cameras deserves a more thorough investigation than is possible in the rebuttal period and is a promising direction for future work.
>
> > Unlike the feature warping approach, the encoded transformation information relies on matrix multiplication. To be honest, I don't fully grasp the physical meaning behind this.
>
> We group the channels into multiple 3D points, which encourages the network to learn geometric features.  The attention layers are optimized to match the relevant query and key vectors, therefore each 3-dimensional feature is expected to behave as abstract point clouds. This type of composition of vector spaces is often used in the literature on equivariant neural networks (See Section3 of Deng et al. (2021) or Section 2.6 of Cohen and Welling (2017)) and in the literature on transforming autoencoders (Hinton et al. (2011), Worrall et al . (2017)).

---

> > ### Author Response · Authors · 2023-11-22
> >
> > References
> > * Jianlin Su, Yu Lu, Shengfeng Pan, Ahmed Murtadha, Bo Wen, and Yunfeng Liu. Roformer: Enhanced transformer with rotary position embedding. arXiv.org, 2021.
> > * Geoffrey E Hinton, Alex Krizhevsky, and Sida D Wang. Transforming auto-encoders. In Proc. of the International Conf. on Artificial Neural Networks (ICANN), pp. 44–51, 2011
> > * Daniel E Worrall, Stephan J Garbin, Daniyar Turmukhambetov, and Gabriel J Brostow. Interpretable transformations with encoder-decoder networks. In Proc. of the IEEE International Conf. on Computer Vision (ICCV), 2017.
> > * Congyue Deng, Or Litany, Yueqi Duan, Adrien Poulenard, Andrea Tagliasacchi, and Leonidas Guibas. Vector neurons: A general framework for so (3)-equivariant networks. In Proc. of the IEEE International Conf. on Computer Vision (ICCV), 2021.
> > * Taco S. Cohen, and Max Welling. Steerable CNNs.  In Proc. of the International Conf. on Learning Representations (ICLR), 2017

---

> > ### Comment · Reviewer_ZuJ5 · 2023-11-23
> >
> > Thank you for the thorough response. Most of my questions have been addressed.  I will maintain the existing score.

---

### Official Review · Reviewer_myD9 · 2023-10-31

**Soundness:** 3 good
**Presentation:** 4 excellent
**Contribution:** 3 good
**Rating:** 8
**Confidence:** 3

**Summary:**

This work proposes an attention mechanism for multi-view transformers which factors in the geometry information, such as camera pose and image position, of the inputs. This differs from most previous work that often condition Transformers using absolute positional encoding of that information. A key intuition given by the authors is that attention from one token to another should happen in a canonical coordinate space.
Sparse-view novel-view synethesis results on a number of datasets show that the proposed method is state-of-the-art compared against other recent methods as well as a number of baselines. These datasets include synthetic yet challenging scenes (like MSN-Hard) as well as real world indoor and outdoor scenes (ACID and RealEstate10k).

**Strengths:**

This is a very well written and for the most part clear paper, with a clear motivation and intuition behind their method, as well as a proper literature review of relevant prior works.

It aims to tackle an important and challenging problem of novel view synthesis in the sparse (few input images) and wide-baseline (long-range camera variation) case; as such, strong results in challenging benchmarks imply a significant contribution to the community.

The method and claims are sound. The method itself does not involve a big departure from the standard Transformer architecture, which means it's easy to implement and widely applicably potentially to other problems.

The experimental set-up is generally thorough. A number of SoTA method are included for comparison and different synthetic as well as real-world datasets are used for evaluation.

**Weaknesses:**

While I agree with the intuition behind this method, what the actual implementation does, and how it affects attention, is somewhat unclear to me. It makes sense to transform features of different views to canonical reference space during attention, and that's obvious only when those are actual coordinates (e.g. points in 3D space); it's not so obvious when features are linear projections of other features.


It's also not obvious that block concatenation of multiple group representations, each with a certain multiplicity, is the best way to achieve it, and I believe this work could do with a bit more discussion on these points, and potentially further ablations/baselines of other ways one could implement the same intuition.

**Questions:**

A number of questions/suggestions:

- As suggested above, I would like to see a variant of GTA that does not involve block concatenation of the geometry representations matrices along with their multiplicites (to mnatch the transformation to the input dimensionality `d`). Have the authors explored a variant such as: concatenate the flattened representation of all groups, followed by a linear projection up to `d`, element-wise multplication with input features (with the rest of the GTA method as proposed)?

- The definition of $\rho_{g_i g_j}$ in Eq. 4 I believe is missing, which makes it difficult to understand how the authors go from Eq 4. to Eq. 5, as well as their claims regarding the quadratic complexity in Eq 4.

- While some results are provided on real world datasets, these are fairly limited in both size and quality (from looking at the videos). It's therefore difficult to conclude that this method is SoTA more generally in real-world cases. For instance, results would be strengthened if a study of the sensitivity of the model's predictions are to camera errors during training and/or test time. This is relevant given that other methods in the literature have carried out these types of experiments (see e.g. SRT paper, Fig 6).

---

> ### Author Response · Authors · 2023-11-22
> **Response to reviewer myD9**
>
> Thank you very much for your feedback and suggestions. We respond to each of your concerns one by one below:
>
> >While I agree with the intuition behind this method, what the actual implementation does, and how it affects attention, is somewhat unclear to me. It makes sense to transform features of different views to canonical reference space during attention, and that's obvious only when those are actual coordinates (e.g. points in 3D space); it's not so obvious when features are linear projections of other features.
>
> We treated the channels into multiple 3D points, which encourages the network to learn geometric features. This composition of vector spaces is often used in the literature on equivariant neural networks (See Section3 of Deng et al. (2021) or Section 2.6 of Cohen and Welling (2017)) and in the literature on transforming autoencoders (Hinton et al. (2011), Worrall et al. (2017)).
>
> We would like to mention that one potential advantage of integrating geometric information as group representations lies in the structured nature of these representations, i.e. the homomorphism property of the representations. In our approach, each QKV vector is transformed by ρ(g) in the same way irrespective of its value, ensuring consistency of the transformed vectors. This makes it easier for the subsequent layers to understand the effect of the transformations. Conversely, with absolute positional encoding (APE) and relative positional encoding (RPE), where the flattened camera transformations are added directly to the input, challenges arise. Not only does the flattened matrix lose its geometric structure when merged with each $Q$, $K$, and $V$, but subsequent layers are also burdened with learning the effect of complicated 3D transformations.
>
>
> >It's also not obvious that block concatenation of multiple group representations, each with a certain multiplicity, is the best way to achieve it, and I believe this work could do with a bit more discussion on these points, and potentially further ablations/baselines of other ways one could implement the same intuition.
>
> Thank you for your suggestions. To address your concern, we additionally conducted experiments with Kronecker product-based GTA we describe below.  There are two conventional ways to construct representation matrices in representation theory: 1. the direct sum and 2. the Kronecker product of irreducible representations. The former is one we described in sections 3.1 and 3.2.
>
> The Kronecker product of two square matrices $A, B \in \mathbb{R}^{m\times m}, \mathbb{R}^{n\times n}$ is defined as
>
> $
> A\otimes B = \begin{bmatrix}
>     a_{11} B & \cdots & a_{1m} B \\\  \vdots  & \ddots  &\vdots \\\  a_{m1}B & \cdots  & a_{mm} B
> \end{bmatrix} \in \mathbb{R}^{mn\times mn}
> $
>
> The important property of the Kronecker product is that the Kronecker product of two representations is also a representation (homomorphism):  $(\rho_1 \otimes \rho_2)(gg') = (\rho_1 \otimes \rho_2)(g) (\rho_1 \otimes \rho_2)(g') $ where $(\rho_1\otimes \rho_2)(g) := \rho_1(g) \otimes \rho_2(g)$.
> We additionally experimented with this Kronecker version of GTA on CLEVR-TR and MSN-Hard, where we use the Kronecker product of the $SE(3)$ representation and $SO(2)$ representations for $\rho_g$ in GTA:
>
> \begin{align}
>     \rho_g = \rho_{\rm cam} (c) \otimes (\rho_{h} (\theta_h) \oplus \rho_{w} (\theta_w)), \text{where $g = (c, \theta_h, \theta_w)$}.
> \end{align}
>
> The multiplicity of $\rho_{\rm cam}$, $\rho_{h}$, $\rho_{g}$ are set to 1, and the number of frequencies for both $\rho_{h}$ and $\rho_{w}$ is set to 4 on CLEVR-TR and 6 on MSN-Hard.
> Below is the comparison between the Kronecker version of GTA (GTA-Kronecker) with the GTA described. Because both GTA-Kronecker and the original GTA encode the geometric structure in a homomorphic way, the performance is relatively close to each other, especially on MSN-Hard.  We added these results to Table 6 in the Appendix.
> | Method           	|  CLEVR-TR PSNR  | MSN-Hard PSNR |
> |----------------------|:---------------:|:-------------:|
> | GTA-kronekcer (ours) |   	38.32    	| 	24.52 	|
> | GTA (ours)       	|	**38.99**	|   **24.58**   |

---

> ### Author Response · Authors · 2023-11-22
>
> >I would like to see a variant of GTA that does not involve block concatenation of the geometry representations matrices along with their multiplicites (to mnatch the transformation to the input dimensionality d). Have the authors explored a variant such as: concatenate the flattened representation of all groups, followed by a linear projection up to d, element-wise multplication with input features (with the rest of the GTA method as proposed)?
>
> Thank you for your question and suggestion of an interesting variant of GTA. We had not tested element-wise multiplication of linearly transformed flattened representations and additionally ran an experiment with that approach. Note that when we transform the representations into vector form multiplied by QKV vectors, we lose the homomorphism property: suppose $f(g) :=W vec(\rho(g))$ is the vector embedding of the representation $\rho(g)$ where $vec()$ is the flattening operation, then $f(gg’) \neq f(g)f(g’)$ in general. Thus we could say f is not faithful in terms of structure preservation of $g$. The table below shows that the performance significantly degrades with element-wise multiplication to QKV features (and outputs of QKV attn.).
>
> | Method|  CLEVR-TR PSNR  |
> |-----------------------------|:---------------:|
> | Element-wise multiplication |  	34.33   	|
> | GTA (ours) |	**38.99**	|
>
> We included this experimental result in Table. 5.
>
>
> >For instance, results would be strengthened if a study of the sensitivity of the model's predictions are to camera errors during training and/or test time.
>
> Thank you for your suggestion. We additionally train models with camera noise added to input views, which is the same setting as in the SRT paper. All modes are trained with σ = 0.1 and tested with different noise strengths on the CLEVR-TR dataset. We describe the experimental settings in detail in Appendix B.4.
> The results are shown below. GTA shows better performance than RePAST regardless of the noise level.  The table below was added as Table 9. in the revised manuscript.
> | σ                      			|  0.01   |	0.1 	|
> |----------------------------- |:-------:|:----------:|
> | (RePAST, wo/noise during training) |  33.80  |  30.57  |
> | RePAST (Safin et al. 2023)  	|  35.26  | 35.14 |
> | GTA (ours) |  **36.66**  |  **36.57**  |
>
> Because of the limited time of the discussion period, we could not train on MSN-Hard for this experiment,
> but will add results on that dataset in the final version.
> Note that, RealEstate10k and ACID datasets include camera noise because each camera parameter is estimated with a structure from motion algorithm. We see even in those settings, the model outperforms Du et al. 2023, which uses APE-based positional encoding.
>
> >The definition of ρ_{g_ig_j} in Eq. 4 I believe is missing, which makes it difficult to understand how the authors go from Eq 4. to Eq. 5, as well as their claims regarding the quadratic complexity in Eq 4.
>
> Thank you for pointing this out. $\rho_{g_i g_j^{-1}} = \rho(g_i g_j^{-1}) \in \mathbb{R}^{d\times d}$ is a representation of the relative group element $g_i g_j^{-1}$ between $i$ and $j$-th tokens. From the homomorphism property of ρ, this can be decomposed into the product of the representations of each group element:  $\rho(g_i g_j^{-1}) =  \rho(g_i) \rho(g_j^{-1})$, with which we can derive the equation (5). (An example of the homomorphism: suppose $g$ is a degree of 2D rotation and $\rho(g): R(g)$ is a 2D rotation matrix given a degree $g$. Then for example, we have $R(40) = R(20+20) = R(20)R(20))$.
> In the revised manuscript, we explicitly mention the homomorphism property after Eq. 4 and define the homomorphism property when we introduce group and representation in Section 3.
>
>
> References:
> * Geoffrey E Hinton, Alex Krizhevsky, and Sida D Wang. Transforming auto-encoders. In Proc. of the International Conf. on Artificial Neural Networks (ICANN), pp. 44–51, 2011
> * Daniel E Worrall, Stephan J Garbin, Daniyar Turmukhambetov, and Gabriel J Brostow. Interpretable transformations with encoder-decoder networks. In Proc. of the IEEE International Conf. on Computer Vision (ICCV), 2017.
> * Congyue Deng, Or Litany, Yueqi Duan, Adrien Poulenard, Andrea Tagliasacchi, and Leonidas Guibas. Vector neurons: A general framework for so (3)-equivariant networks. In Proc. of the IEEE International Conf. on Computer Vision (ICCV), 2021.
> * Taco S. Cohen, and Max Welling. Steerable CNNs.  In Proc. of the International Conf. on Learning Representations (ICLR), 2017

---

### Author Response · Authors · 2023-11-22
**General responses**

We thank all reviewers for their valuable feedback. All reviews agree that our paper presents a well-motivated method with strong performance and thorough experiments.
Below we show two main experiments we additionally conducted. All results were added to our new manuscript.
We highlighted new parts in blue in the revised manuscript.

As suggested, we added a comparison of additional 3D positional encoding schemes on the CLEVR-TR dataset, including methods that were not originally proposed for the novel view synthesis task.

| Method                                    	| PSNR  	|
|-----------------------------------------------|-----------|
| MLN (Liu et al. 2023)                     	| 31.97   	|
| Element-wise Mul.                         	| 34.33   	|
| GBT (Venkat et al. 2023)                  | 35.63 |
| Frustum Embedding (Liu et al. 2022)   | 37.23 |
| RoPE+FTL (Su et al. 2021, Worrall et al. 2017) | 38.18   	|
| GTA                                       	| **38.99** |


* Frustum Embedding (Liu et al. 2022): Positional encoding scheme of the PETR object detection method. 3D points at multiple depths are generated along every pixel, encoded with an MLP, and added to the encoded tokens.
* MLN (Liu et al. 2023): modulates and biases tokens by using linearly transformed flattened representation matrices.
* Elementwise Mul.: instead of the representation matrix used in GTA, uses a vector obtained from flattened representation matrices and multiplies it with QKV features element-wise.
* GBT (Venkat et al. 2023): bias the attention matrix by using the ray distance between query and key tokens.
* RoPE (Su et al. 2021)+FTL (Worrall et al. 2017): similar to GTA but removes SE(3) representations from attention layers, and applies the SE(3) transformation only to the encoder output.

We see that GTA achieves better view reconstruction quality than other positional encoding schemes. We added these results to Table 5 in Section 4 in the revised manuscript. Experimental settings for this comparison are described in the Appendix E.4.

Also suggested by reviewer LyNT, we restructured our manuscript and added more discussions of positional encoding schemes from other 3D vision fields (Frustum posemb, MLN, etc…), as well as, geometry-aware attention mechanisms in NVS (GBT, Epipolar sampling attention).

As suggested by reviewer nfN9 we added additional NeRF-based prior work on RealEstate10k to Table 3. Additionally, we trained and tested the latest concurrent NeRF-based approach MatchNeRF from arXiv on that dataset. We find that MatchNeRF outperforms both NeRF-based and transformer-based prior work by large margins.
By combining our GTA mechanism with Du et al. 2023, the transformer-based method also achieves state-of-the-art results. We would like to mention that the transformer-based NVS with GTA, which introduces a relatively weaker prior into the model, is competitive with the model equipped with a strong physical prior (volumetric rendering).  Our attention mechanism effectively steers the transformer to learn the underlying 3D information behind images.

|                                    	| RealEstate10k |  |  | ACID  |  |  |
|----------------------------------------|-------|-------|-------|-------|-------|-------|
| Model                              	| PSNR  | LPIPS  | SSIM   | PSNR  | LPIPS  | SSIM  |
| PixelNeRF (Yu et al. 2021)         	| 13.91 | 0.591 | 0.460 | 16.48 | 0.628 | 0.464 |
| StereoNeRF (Chibane et al. 2021)   	| 15.40 | 0.604 | 0.486 |  -	|  -	|  -	|
| IBRNet (Wang et al. 2021)          	| 15.99 | 0.532 | 0.484 | 19.24 | 0.385 | 0.513 |
| GeoNeRF (Johari et al. 2022)       	| 16.65 | 0.541 | 0.511 |  -	|  -	|  -	|
| MatchNeRF (Chen et al. 2023)       	| **23.06** | 0.258 | 0.830 |  -	|  -	|  -	|
| -------------------------------------- |-------|-------|-------|-------|-------|-------|
| GPNR (Suhail et al. 2022)          	| 18.55 | 0.459 | 0.748 | 17.57 | 0.558 | 0.719 |
| Du et al. 2023               	| 21.65 | 0.285 | 0.822 | 23.35 | 0.334 | 0.801 |
| Du et al. 2023 + GTA (Ours)          | 22.85 | **0.255** | **0.850** |  **24.10** |   **0.291** | **0.824** |

---

> ### Author Response · Authors · 2023-11-22
> **References**
>
> * Yingfei Liu, Tiancai Wang, Xiangyu Zhang, and Jian Sun. Petr: Position embedding transformation for multiview 3d object detection. In Proc. of the European Conf. on Computer Vision (ECCV), 2022.
> * Ruoshi Liu, Rundi Wu, Basile Van Hoorick, Pavel Tokmakov, Sergey Zakharov, and Carl Vondrick. Zero-1-to-3: Zero-shot one image to 3d object. arXiv.org, 2023.
> * Naveen Venkat, Mayank Agarwal, Maneesh Singh, and Shubham Tulsiani. Geometry-biased transformers for novel view synthesis. arXiv.org, 2023.
> * Jianlin Su, Yu Lu, Shengfeng Pan, Ahmed Murtadha, Bo Wen, and Yunfeng Liu. Roformer: Enhanced transformer with rotary position embedding. arXiv.org, 2021.
> * Daniel E Worrall, Stephan J Garbin, Daniyar Turmukhambetov, and Gabriel J Brostow. Interpretable transformations with encoder-decoder networks. In Proc. of the IEEE International Conf. on Computer Vision (ICCV), 2017.
> * Alex Yu, Vickie Ye, Matthew Tancik, and Angjoo Kanazawa. PixelNeRF: Neural radiance fields from one or few images. In Proc. IEEE Conf. on Computer Vision and Pattern Recognition (CVPR), 2021
> * Julian Chibane, Aayush Bansal, Verica Lazova, and Gerard Pons-Moll. Stereo radiance fields (SRF): learning view synthesis for sparse views of novel scenes. In Proc. IEEE Conf. on Computer Vision and Pattern Recognition (CVPR), 2021.
> * Qianqian Wang, Zhicheng Wang, Kyle Genova, Pratul P Srinivasan, Howard Zhou, Jonathan T Barron, Ricardo Martin-Brualla, Noah Snavely, and Thomas Funkhouser. IBRNet: Learning multi-view image-based rendering. In Proc. IEEE Conf. on Computer Vision and Pattern Recognition (CVPR), 2021
> * Mohammad Mahdi Johari, Yann Lepoittevin, and Franc ̧ois Fleuret. Geonerf: Generalizing nerf with geometry priors. In Proc. IEEE Conf. on Computer Vision and Pattern Recognition (CVPR), 2022.
> * Yuedong Chen, Haofei Xu, Qianyi Wu, Chuanxia Zheng, Tat-Jen Cham, and Jianfei Cai. Explicit correspondence matching for generalizable neural radiance fields. arXiv.org, 2023.
> * Yilun Du, Cameron Smith, Ayush Tewari, and Vincent Sitzmann. Learning to render novel views from wide-baseline stereo pairs. In Proc. IEEE Conf. on Computer Vision and Pattern Recognition (CVPR), 2023

---

### Meta-Review · Area_Chair_YV8j · 2023-12-11

**Metareview:**

This paper presents a new approach to enhancing attention in 3D vision tasks by incorporating geometric information. The paper received positive ratings: 8, 6, 6, 5, and all reviewers acknowledged the paper's technical contribution and its superior performance compared to other leading methods. There were concerns about the novelty of the method and the absence of extensive comparisons and analysis of its components. The authors properly addressed these concerns in their response, conducting additional experiments, and providing more clarifications. In conclusion, the reviewers and AC believe that the paper introduces a promising approach to 3D vision tasks and recommend accepting the paper.

**Justification For Why Not Higher Score:**

The decision to not assign a higher score is influenced by the paper's need for further improvement and refinement to become stronger. Additionally, the level of novelty presented in the paper, while noteworthy, is not exceptional.

**Justification For Why Not Lower Score:**

The reviewers and AC believe that the paper introduces a promising approach to 3D vision tasks and recommend accepting the paper.

---

### Decision · Program_Chairs · 2024-01-16

Accept (poster)